# GSRF: Complex-Valued 3D Gaussian Splatting for Efficient Radio-Frequency Data Synthesis

**Kang Yang**[1†]    **Gaofeng Dong**[1]    **Sijie Ji**[1,2]    **Wan Du**[3✉‡]    **Mani Srivastava**[1§]

[1]University of California, Los Angeles    [2]California Institute of Technology
[3]University of California, Merced

{kyang73, gfdong}@g.ucla.edu    sijieji@caltech.edu
wdu3@ucmerced.edu[✉]    mbs@ucla.edu

## Abstract

Synthesizing radio-frequency (RF) data given the transmitter and receiver positions, *e.g.*, received signal strength indicator (RSSI), is critical for wireless networking and sensing applications, such as indoor localization. However, it remains challenging due to complex propagation interactions, including reflection, diffraction, and scattering. State-of-the-art neural radiance field (NeRF)-based methods achieve high-fidelity RF data synthesis but are limited by long training times and high inference latency. We introduce GSRF, a framework that extends 3D Gaussian Splatting (3DGS) from the optical domain to the RF domain, enabling efficient RF data synthesis. GSRF realizes this adaptation through three key innovations: First, it introduces complex-valued 3D Gaussians with a hybrid Fourier–Legendre basis to model directional and phase-dependent radiance. Second, it employs orthographic splatting for efficient ray–Gaussian intersection identification. Third, it incorporates a complex-valued ray tracing algorithm, executed on RF-customized CUDA kernels and grounded in wavefront propagation principles, to synthesize RF data in real time. Evaluated across various RF technologies, GSRF preserves high-fidelity RF data synthesis while achieving significant improvements in training efficiency, shorter training time, and reduced inference latency.

## 1 Introduction

Wireless networks, *e.g.*, WiFi and Fifth Generation (5G) cellular networks, are increasingly tasked with supporting both communication and sensing applications through deep learning (DL) models, including indoor localization [1, 2, 3]. However, training these DL models requires large-scale radio-frequency (RF) datasets, *e.g.*, received signal strength indicator (RSSI) measurements across different transmitter and receiver positions within a 3D space, which are typically collected through site surveys. These site surveys involve labor-intensive and time-consuming RF signal measurements across numerous transmitter–receiver locations [4, 5, 6].

Inspired by the success of generative models in computer vision [7, 8, 9, 10], a natural alternative approach is to synthesize RF data through propagation modeling, which computes the received RF signal at a receiver given a transmitter emitting signals from a specific position [11]. However, generating high-fidelity RF data is challenging due to complex propagation interactions between RF signals and surrounding objects, including reflection, diffraction, and scattering.

---

[†]This work was partially done when Kang Yang was a PhD student in Dr. Wan Du's group at UC Merced.

[‡]Corresponding Author: wdu3@ucmerced.edu

[§]Mani Srivastava holds concurrent appointments as a Professor of ECE and CS (joint) at the University of California, Los Angeles, and as an Amazon Scholar at Amazon. This paper describes work performed at UCLA and is not associated with Amazon.

39th Conference on Neural Information Processing Systems (NeurIPS 2025).

Neural Radiance Field (NeRF) [7]-based methods [12, 13] address these challenges by extending NeRF to the RF domain, achieving state-of-the-art fidelity in RF data synthesis. These NeRF-based methods adopt continuous RF scene representations to effectively model complex RF interactions. However, their stochastic sampling process and Multi-layer Perceptron (MLP) optimization are computationally intensive and slow, limiting real-time applicability. Efficient training and inference in the RF domain are crucial for applications such as real-time localization and tracking [14, 6].

This paper proposes GSRF, an efficient RF data synthesis framework that extends 3D Gaussian Splatting (3DGS) [8, 15], developed for real-time novel view synthesis, to the RF domain. However, this adaptation introduces challenges due to inherent differences between visible light and RF signals:

• *(i) Directional and Phase Modeling.* In 3DGS [8, 16], the color attribute of a Gaussian distribution is parameterized by spherical harmonics (SH) coefficients [17, 8] to capture directional variations caused by optical propagation effects such as reflections and shading. In contrast, RF signals with centimeter-scale wavelengths exhibit complex phenomena such as diffraction [18] and phase-dependent interference (constructive and destructive), which SH coefficients struggle to capture [19].

• *(ii) Data Capture Mechanism.* In visible light, images are captured by camera sensors (*e.g.*, CMOS or CCD) on a 2D image plane, allowing splatting through classical transformation matrices that project 3D Gaussians onto the plane to identify ray-Gaussian intersections. In contrast, RF signals are collected by antenna arrays over a spherical region centered at the RF antenna. This fundamental difference makes splatting algorithms designed for visible light unsuitable for the RF domain.

• *(iii) Rendering Algorithm.* In 3DGS, point-based rendering algorithm aggregates amplitude-based attributes, *e.g.*, color, to compute pixel values along each ray. In contrast, RF signal synthesis needs to consider both amplitude and phase to model interference patterns. This necessitates a complex-valued rendering algorithm, along with CUDA kernels that jointly process amplitude and phase information.

By tackling the three challenges above, we make the following key contributions:

• *Fourier-Legendre Radiance Fields.* A scene is represented using 3D Gaussian distributions, each characterized by four attributes: a mean and covariance matrix, along with two RF-specific attributes, which are complex-valued RF radiance and transmittance. The directional radiance is modeled using a Fourier-Legendre Expansion (FLE) [20]. FLE leverages Fourier basis functions for the azimuthal angle $\alpha$ and Legendre polynomials for the elevation angle $\beta$, with complex coefficients $c_{ml} \in \mathbb{C}$ encoding both amplitude and phase. Additionally, the complex-valued transmittance models signal amplitude attenuation and phase shifts as the RF signal propagates through a Gaussian.

• *Orthographic Splatting.* To determine ray-Gaussian intersections, GSRF introduces an orthographic splatting method for the RF domain. GSRF operates on the Ray Emitting Spherical Surface (RESS), a spherical region where RF signals are captured. Each 3D Gaussian is then splatted onto this region via orthographic projection, enabling identification of intersecting Gaussians for each ray.

• *Complex-Valued Ray Tracing.* GSRF incorporates a complex-valued ray tracing algorithm for RF signals, executed on RF-customized CUDA kernels. Building on the Huygens-Fresnel principle [21], which states that each point on a wavefront acts as a source of secondary wavelets, GSRF models each Gaussian as an RF source. GSRF emits rays from the RESS, identifies intersecting Gaussians through the adapted splatting method, and employs a complex-valued ray tracing algorithm to jointly process amplitude and phase attributes along each ray, computing the received RF signal data.

• GSRF is trained with an RF-customized loss function derived from both time and frequency domains using 2D Fourier transforms to capture the intricate propagation characteristics of RF signals.

We evaluate GSRF on various RF technologies, including radio-frequency identification (RFID), Bluetooth Low Energy (BLE), and 5G networks, to synthesize different types of RF data, including RSSI, spatial spectra, and complex-valued channel state information (CSI). Results show that GSRF achieves significantly higher efficiency than existing methods, with improvements in training data efficiency, training time, and inference latency. We release our code at this GitHub repository.

## 2 Preliminaries

**RF Signal Propagation Characteristics.** Wireless systems, such as WiFi, rely on RF signals propagating between transmitters and receivers [22, 23]. A transmitted signal can be represented as:

$$s(t) = Ae^{j(2\pi f_c t + \theta)}, \tag{1}$$

where $A$ is the amplitude, $f_c$ is the carrier frequency (*e.g.*, 2.4 GHz), and $\theta$ is the initial phase. As the signal propagates through the scene, it encounters obstacles that cause reflections, diffraction, and scattering, resulting in multiple propagation paths. The received signal is the sum of these paths:

$$r(t) = \sum_{i=1}^{N} A_i e^{j\phi_i} s\left(t - \tau_i\right), \quad \phi_i = 2\pi f_c \tau_i + \theta_i, \quad \tau_i = \frac{d_i}{c} \tag{2}$$

where $N$ is the number of paths, $c$ is the RF signal speed, $A_i$ is the attenuated amplitude, $\phi_i$ is the phase shift, $\tau_i$ is the time delay of path length $d_i$, and $\theta_i$ is the phase change from reflections.

The phase greatly affects the received signal, as illustrated in the following example. For two paths with lengths $d_1 = 3\,\text{m}$ and $d_2 = 3.0625\,\text{m}$ at a carrier frequency of $f_c = 2.4\,\text{GHz}$, the corresponding delays are $\tau_1 = 10\,\text{ns}$ and $\tau_2 = 10.208\,\text{ns}$. The phase shifts are $\phi_1 = 0$ and $\phi_2 = \pi$, resulting in a phase difference of $\Delta\phi = \pi$, which causes destructive interference, *i.e.*, the two signals cancel each other out, leading to a reduction or complete loss of signal strength. Conversely, when $\Delta\phi \approx 0$, constructive interference occurs, amplifying the signal [22]. Therefore, synthesizing RF data requires modeling these amplitude and phase interactions across all paths.

**3D Gaussian Splatting (3DGS).** It is a real-time rendering technique for novel view synthesis in 3D scenes [8]. It represents a 3D scene as a collection of 3D Gaussian ellipsoids $\{\zeta_1, \ldots, \zeta_K\}$, where each Gaussian primitive $\zeta_k$ is defined by a 3D Gaussian distribution:

$$G_k\left(x; \mu_k, \Sigma_k\right) = \exp\left(-\frac{1}{2}\left(x - \mu_k\right)^\mathsf{T} \Sigma_k^{-1}\left(x - \mu_k\right)\right), \tag{3}$$

where $\mu_k \in \mathbb{R}^3$ is the center position and $\Sigma_k \in \mathbb{R}^{3\times3}$ is the covariance matrix. It is decomposed as: $\Sigma_k = R_k S_k S_k^\mathsf{T} R_k^\mathsf{T}$, where $R_k$ and $S_k$ are learnable rotation and scaling matrices that ensure positive semi-definiteness [8]. Each Gaussian also includes an opacity term $\rho_k \in [0, 1]$ and SH coefficients $sh_k \in \mathbb{R}^d$, making each Gaussian primitive represented as: $\zeta_k = (\mu_k, R_k, S_k, \rho_k, sh_k)$.

To render an image, 3DGS projects these 3D Gaussians onto a 2D image plane, forming 2D Gaussians:

$$G_k^{2D}\left(r; \mu_k^{2D}, \Sigma_k^{2D}\right) \quad \text{with} \quad \mu_k^{2D} = \pi_{2D}\left(\mu_k\right), \quad \Sigma_k^{2D} = JW\Sigma_k W^\mathsf{T} J^\mathsf{T}, \tag{4}$$

where $J$ is the Jacobian of the projective transformation, and $W$ is the world-to-camera transformation matrix [8]. The pixel color $\hat{C}(r)$ at location $r \in \mathbb{R}^2$ is computed via $\alpha$-blending:

$$\hat{C}(r) = \sum_{k \in \mathcal{S}_r} \omega_k^{2D}(r)\, c\left(sh_k, r\right), \tag{5}$$

where $\mathcal{S}_r \subseteq \{1, \ldots, K\}$ is the subset of indices of Gaussians that contribute to pixel $r$. The term $\omega_k^{2D}(r)$ represents the contribution of each Gaussian, computed as:

$$\omega_k^{2D}(r) = \rho_k G_k^{2D}\left(r; \mu_k^{2D}, \Sigma_k^{2D}\right) \prod_{j=1}^{k-1}\left(1 - \rho_j G_j^{2D}\left(r; \mu_j^{2D}, \Sigma_j^{2D}\right)\right), \tag{6}$$

where the Gaussians are ordered by increasing depth (*i.e.*, from front to back) to ensure correct rendering. Finally, $c\left(sh_k, r\right)$ is the color decoded from the SH coefficients $sh_k$.

## 3 Related Work

Conventional RF data synthesis methods include simulations [24, 25, 26], empirical models [22, 27, 28], and physics-unaware DL models [29, 30, 31], but all suffer from low modeling fidelity due to inherent limitations. Simulations require accurate scene Computer-Aided Design (CAD) models, which are often unavailable. Empirical models oversimplify propagation with limited parameters, predicting only coarse signal power. Physics-unaware DL models map inputs to labels but fail to capture the underlying physics of RF propagation. NeRF-based methods [12, 13, 32] introduce voxel-based scene representations to capture scene impact on RF signal propagation and employ ray tracing algorithms to achieve state-of-the-art fidelity in RF data synthesis. However, they suffer from low efficiency, requiring long training times and exhibiting high inference latency. This work proposes a 3DGS-based method to achieve high training and inference efficiency.

Two recent works, RF-3DGS [33] and WRF-GS [34], propose 3DGS-inspired techniques for RF data synthesis, yet both face limitations. RF-3DGS [33] employs a two-stage training process to

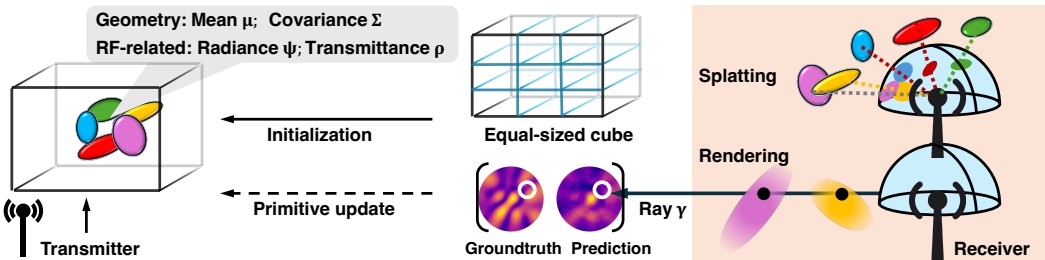

Figure 1: Overview of GSRF architecture. The RF scene is represented by Gaussian primitives with mean $\mu$, covariance $\Sigma$, and complex-valued radiance $\psi$ and transmittance $\rho$, whose attributes are updated via gradient-based optimization with adaptive density control. For rendering, rays $\gamma$ are emitted from the receiver, Gaussians are splatted onto a *2D receiving RF plane*, and the received data is obtained by aggregating complex-valued contributions along each ray.

learn scene representations using Gaussian primitives defined by mean, covariance, opacity, and path loss. First, optical 3DGS optimizes mean, covariance, and opacity from visual images, then these parameters are fixed to train path loss with RF data. However, merging visible light and RF signals is challenging due to their distinct properties. Moreover, visual data is often unavailable in RF domains.

WRF-GS [34] assigns each Gaussian four attributes: mean, covariance, radiance, and attenuation. It adopts a NeRF-inspired approach to learn radiance and attenuation by optimizing a large MLP with each Gaussian's position as input. This dependence on a computationally intensive MLP results in inefficiency and introduces NeRF-like bottlenecks: (i) dense querying of the MLPs for attribute prediction during training and inference, (ii) expensive backpropagation through deep networks for every Gaussian update, which scales poorly with scene complexity, and (iii) high inference latency due to per-query MLP evaluations (*e.g.*, for novel transmitter positions). In contrast, GSRF eliminates MLP regressors entirely by directly optimizing per-Gaussian attributes as learnable parameters. Combined with Fourier–Legendre radiance fields, orthographic splatting, and complex-valued ray tracing, this design achieves faster training and inference compared to WRF-GS.

WRF-GS+ is an extension of WRF-GS [34]. It introduces deformable Gaussians that decouple static components (*e.g.*, path loss) and dynamic components (*e.g.*, multipath) via learned offsets, thereby improving synthesis quality and mitigating the inefficiencies of WRF-GS's MLP-based attributes. While effective, this approach remains distinct from GSRF, which offers a unified, complex-valued, MLP-free pipeline for RF propagation modeling; nevertheless, deformable mechanisms could be explored in future extensions of our framework.

## 4 Methodology

### 4.1 Problem Formulation

Given a transmitter at a fixed position emitting RF signals (*e.g.*, a WiFi router) and a receiver (*e.g.*, a smartphone) distributed throughout a scene, the objective is to synthesize the received RF data. Formally, for a transmitter located at $\mathbf{t} = (x_{\text{tx}}, y_{\text{tx}}, z_{\text{tx}})$ and a set of receiver positions $\{\mathbf{r}_i\}_{i=1}^{N}$, where $\mathbf{r}_i = (x_{\text{rx},i}, y_{\text{rx},i}, z_{\text{rx},i})$, the goal is to estimate a model with parameters $\theta$ that synthesizes the received RF data $S_i$ at each receiver $\mathbf{r}_i$:

$$\theta^* = \operatorname*{argmax}_{\theta} p\left(\{S_i\}_{i=1}^{N} \mid \mathbf{t}, \{\mathbf{r}_i\}_{i=1}^{N}, \theta\right), \tag{7}$$

where $S_i \in \mathbb{C}$ represents the received complex-valued RF data at receiver $\mathbf{r}_i$, encapsulating both amplitude and phase. For specific RF technologies or applications, $S_i$ may represent a scalar signal power $S_i \in \mathbb{R}$ or a spatial spectrum $S_i \in \mathbb{R}^{N_{\text{az}} \times N_{\text{el}}}$ over azimuth $\alpha$ and elevation $\beta$ angles. For example, with a one-degree angular resolution, we have $N_{\text{az}} = 360$ and $N_{\text{el}} = 180$.

**Model Overview.** Figure 1 illustrates the GSRF. First, the scene is represented using 3D Gaussian distributions, each characterized by a mean $\mu_k \in \mathbb{R}^3$, a covariance matrix $\Sigma_k \in \mathbb{R}^{3 \times 3}$, and two RF-specific complex-valued attributes: radiance $\psi_k \in \mathbb{C}$ and transmittance $\rho_k \in \mathbb{C}$. To initialize the mean and covariance matrix, the scene is partitioned into equal-sized cubes, deriving initial scene point clouds without Structure-from-Motion (SfM) algorithms [35, 36], which are inapplicable to the RF domain. Next, each 3D Gaussian is projected onto the receiver's receiving region, using orthographic projection to efficiently identify ray-Gaussian intersections. For each ray, intersecting Gaussians are

sorted by depth, and a complex-valued ray tracing algorithm is applied to compute the received signal. Model optimization is performed by minimizing the loss function. Explicit gradients are computed to update the primitives via stochastic gradient descent, adjusting parameters $(\mu_k, \Sigma_k, \psi_k, \rho_k)$ and refining primitive density through gradient-driven cloning, splitting, or removal.

## 4.2 Fourier-Legendre Radiance Fields

Each Gaussian primitive in GSRF is represented as a tuple:

$$\zeta_k = (\mu_k, R_k, S_k, \psi_k, \rho_k) \quad \text{with} \quad \Sigma_k = R_k S_k S_k^\mathsf{T} R_k^\mathsf{T}. \tag{8}$$

The pair $(\mu_k, R_k, S_k)$ defines a 3D Gaussian distribution resembling an ellipsoid, representing a probability distribution in 3D space. Its probability density function (PDF) is given by Equation (3).

The transmittance $\rho_k \in \mathbb{C}$ models the effect of an RF signal passing through the $k$-th Gaussian, resulting in an amplitude reduction $|\rho_k|$ and a phase shift $\angle \rho_k$. According to Maxwell's equations [37], transmittance depends on the material properties at the Gaussian's location $\mu_k$. Therefore, $\rho_k$ primarily captures the physical interaction of the RF signal with the medium.

The radiance $\psi_k \in \mathbb{C}$ represents the complex-valued RF signal emitted by the $k$-th Gaussian. To model its directional dependency, $\psi_k$ is defined using a Fourier-Legendre Expansion (FLE) [20], which leverages Fourier basis functions for the azimuthal angle $\alpha$ and Legendre polynomials for the elevation angle $\beta$. This approach is physically grounded in the Huygens-Fresnel principle [21], which posits that each point on a wavefront, such as the $k$-th Gaussian at position $\mu_k$, acts as a source of secondary spherical wavelets. The emitted RF signal is modeled as a solution to the wave equation in spherical coordinates, where spherical harmonics—comprising Fourier functions $e^{im\alpha}$ and associated Legendre polynomials $P_l^m(\cos\beta)$—form a complete basis for representing directional wave fields on the unit sphere. Specifically, for a direction $(\alpha, \beta)$, the radiance is expressed as:

$$\psi_k(\alpha, \beta) = \sum_{l=0}^{L} \sum_{m=-l}^{l} c_{ml}^{(k)} e^{im\alpha} P_l^m(\cos\beta), \tag{9}$$

where $c_{ml}^{(k)} \in \mathbb{C}$ are complex coefficients encoding the amplitude and phase of the radiance for the $k$-th Gaussian. This representation effectively captures phase-dependent interference crucial, as the separation of $\alpha$ and $\beta$ aligns with their geometric roles in spherical coordinates, while the complex coefficients model the interference effects stemming from the wave nature of RF signals.

## 4.3 Orthographic Splatting

For a receiver positioned at $\mathbf{r} = (x_{\text{rx}}, y_{\text{rx}}, z_{\text{rx}})$, rays are emitted to sample the RF signal across various directions around the receiver $\mathbf{r}$. Each ray is parameterized as:

$$\gamma(d) = \mathbf{r} + d\hat{\mathbf{v}}, \quad d \geq r_{\text{rx}}, \tag{10}$$

where $d$ is the distance along the ray from the receiver, and $\hat{\mathbf{v}}$ is the unit direction vector. Therefore, rays are emitted from the Ray Emitting Spherical Surface (RESS), which is a sphere centered at $\mathbf{r}$ with radius $r_{\text{rx}}$, and extend outward. For a one-degree angular resolution, $N_{\text{az}} = 360$ and $N_{\text{el}} = 180$, resulting in a total of $360 \times 180$ rays being emitted, covering all directions around the receiver.

**2D Receiving RF Plane.** To enable splatting in the RF domain, where an image plane is absent, we map the RESS onto a 2D RF plane. Consider a point $\mathbf{p} = (x, y, z) \in \mathbb{R}^3$ on the RESS, satisfying $\|\mathbf{p} - \mathbf{r}\| = r_{\text{rx}}$. We transform the Cartesian coordinates of $\mathbf{p}$ into spherical coordinates $(\zeta, \alpha, \beta)$, where $\zeta$ is the radial distance, $\alpha \in [0, 2\pi)$ is the azimuthal angle, and $\beta \in [-\pi/2, \pi/2]$ is the elevation angle:

$$\begin{aligned} \zeta &= \sqrt{x^2 + y^2 + z^2} = r_{\text{rx}}, \\ \alpha &= \arctan 2(y, x), \\ \beta &= \frac{\pi}{2} - \arccos\left(\frac{z}{r_{\text{rx}}}\right). \end{aligned} \tag{11}$$

We then project $\alpha$ and $\beta$ onto a 2D grid with one-degree resolution, defined as:

$$u = \left\lfloor \frac{\alpha \cdot 180}{\pi} \right\rfloor, \quad v = \left\lfloor \frac{\beta \cdot 180}{\pi} + 90 \right\rfloor, \tag{12}$$

where $\lfloor \cdot \rfloor$ denotes the floor function. The resulting coordinates $(u, v)$ define the 2D RF plane.

**Splatting Process.** Each 3D Gaussian, with mean $\mu_k \in \mathbb{R}^3$ and covariance $\Sigma_k \in \mathbb{R}^{3\times3}$, is projected onto the 2D RF plane to identify ray-Gaussian intersections. The unit direction vector from the receiver position $\mathbf{r} \in \mathbb{R}^3$ to the $k$-th Gaussian center $\mu_k$ is given by:

$$\hat{\mathbf{v}}_k = \frac{\mu_k - \mathbf{r}}{\|\mu_k - \mathbf{r}\|_2}. \tag{13}$$

The projected center $\mu_k^{2D} = (u_k, v_k)$ is computed from Equations (11) and (12), with the vector $\hat{\mathbf{v}}_k$ as input. The 3D covariance $\Sigma_k$ is projected onto the 2D plane as $\Sigma_k^{2D} = J\Sigma_k J^{\mathsf{T}}$, where $J$ is the Jacobian matrix, and the 2D spread is approximated by radius $r_k = 3\sqrt{\lambda_{\max}}$, with $\lambda_{\max}$ as the largest eigenvalue of $\Sigma_k^{2D}$. Rays at points $(u, v)$ intersect the Gaussian if: $\sqrt{(u - u_k)^2 + (v - v_k)^2} \leq r_k$.

## 4.4 Complex-Valued Ray Tracing Algorithm

The received signal $S \in \mathbb{C}$ for a ray is computed by aggregating the contributions from all intersecting Gaussians, considering their geometric influence, radiance, and transmittance. The intersecting Gaussians are sorted in ascending order of their distance from the receiver along the ray path to ensure correct accumulation of transmittance effects. The received signal is computed as:

$$S = \sum_{k=1}^{K_{\text{intr}}} \underbrace{\mathcal{G}_k\left(x_{\text{rep},k}; \mu_k, \Sigma_k\right)}_{\text{Gaussian weight}} \cdot \underbrace{\left(|\psi_k| e^{j\angle\psi_k}\right)}_{\text{Complex radiance}} \cdot \underbrace{\prod_{m=1}^{k-1} \left(|\rho_m| e^{j\angle\rho_m}\right)}_{\text{Cumulative transmittance}} \tag{14}$$

where $K_{\text{intr}}$ is the number of Gaussians intersecting the ray, and $\mathcal{G}_k\left(x; \mu_k, \Sigma_k\right)$ is the probability density function of the $k$-th Gaussian, evaluated at the representative intersection point $x_{\text{rep},k} \in \mathbb{R}^3$, which is the midpoint of the intersection points between the ray trajectory and the ellipsoid defined by the Gaussian's mean $\mu_k \in \mathbb{R}^3$ and covariance $\Sigma_k \in \mathbb{R}^{3\times3}$. The term $\mathcal{G}_k\left(x_{\text{rep},k}; \mu_k, \Sigma_k\right)$ weights the radiance based on the Gaussian's density at the intersection point, while the product term accumulates both amplitude attenuation and phase shifts from all preceding Gaussians, capturing both amplitude reduction and phase shifts during propagation. The detail of proor is provided in Appendix A.

**Loss Function.** The loss function is designed based on the receiver antenna type.

ANTENNA ARRAY. For a receiver equipped with antenna arrays, the signal power across all directions is represented as a ground-truth spatial spectrum matrix $\mathbf{S} \in \mathbb{R}^{N_{\text{az}} \times N_{\text{el}}}$, spanning $N_{\text{az}}$ azimuth and $N_{\text{el}}$ elevation angles. The predicted spatial spectrum is denoted as $\hat{\mathbf{S}} \in \mathbb{R}^{N_{\text{az}} \times N_{\text{el}}}$. The loss function $\mathcal{L}$ combines the $\mathcal{L}_1$ loss, the Structural Similarity Index Measure (SSIM) loss, and a Fourier-based loss:

$$\mathcal{L} = (1 - \lambda_1 - \lambda_2)\mathcal{L}_1 + \lambda_1 \mathcal{L}_{\text{SSIM}} + \lambda_2 \mathcal{L}_{\text{Fourier}}, \tag{15}$$

where $\mathcal{L}_1 = \frac{1}{N_{\text{az}}N_{\text{el}}} \sum_{u=1}^{N_{\text{az}}} \sum_{v=1}^{N_{\text{el}}} \left|\hat{\mathbf{S}}(u, v) - \mathbf{S}(u, v)\right|$ measures the average absolute difference between the predicted and ground-truth spectra. The term $\mathcal{L}_{\text{SSIM}}$ captures spatial RF pattern similarity across directions. The term $\mathcal{L}_{\text{Fourier}}$ quantifies the difference in the frequency domain:

$$\mathcal{L}_{\text{Fourier}} = \frac{1}{N_{\text{az}}N_{\text{el}}} \sum_{f_u, f_v} \left|\mathcal{F}\left(\hat{\mathbf{S}}\right)(f_u, f_v) - \mathcal{F}(\mathbf{S})(f_u, f_v)\right|^2, \tag{16}$$

where $\mathcal{F}\left(\hat{\mathbf{S}}\right)(f_u, f_v)$ and $\mathcal{F}(\mathbf{S})(f_u, f_v)$ are the 2D Fourier transforms of the predicted and ground-truth spectra, respectively, with $f_u$ and $f_v$ representing the frequency indices in the azimuth and elevation dimensions. This term promotes consistency in the frequency domain, which is important for learning RF propagation behavior. The squared magnitude penalizes discrepancies in both amplitude and phase, enhancing the fidelity of synthesized signals.

SINGLE ANTENNA. For a receiver equipped with a single antenna, the ground-truth received signal $S$ represents either a real-valued power measurement or a complex-valued signal encompassing both amplitude and phase information. The synthesized signal $\hat{S}$ is computed as $\hat{S} = \sum_{u=1}^{N_{\text{az}}} \sum_{v=1}^{N_{\text{el}}} \hat{S}_{u,v}$, where $\hat{S}_{u,v}$ denotes the synthesized signal contribution from the ray at azimuth index $u \in \{1, \ldots, N_{\text{az}}\}$ and elevation index $v \in \{1, \ldots, N_{\text{el}}\}$. The loss function $\mathcal{L}$ is defined as $\mathcal{L} = \left\|\hat{S} - S\right\|_1$ if $S$ is real-valued (RF signal power), and as $\mathcal{L} = \left\|\hat{S} - S\right\|_2^2$ if $S$ is complex-valued, penalizing both amplitude and phase errors.

**Gradient-Based Gaussian Primitive Optimization.** GSRF initializes the number of Gaussians and their primitives based on the scene's point clouds, which are obtained by partitioning the scene into equal-sized cubes. After calculating the loss function, the optimization of Gaussian primitives is performed through gradient-based strategies, as detailed in Appendix B.

**Fast Differentiable RF Signal Renderer for Gaussians.** In GSRF, we develop two CUDA kernels to enable efficient forward and backward computations for differentiable RF signal synthesis using Gaussian primitives. Implementation details of the CUDA kernels are provided in Appendix C. To reduce computational overhead, gradients are explicitly calculated as described in Appendix D.

## 5 Experiments

Our method is implemented in PyTorch with CUDA. Further implementation details and hyperparameter settings are provided in Appendix E, and additional experiments are presented in Appendix F.

We evaluate GSRF across three RF technologies for various RF data synthesis tasks: • *(i) Radio-Frequency Identification (RFID)* for spatial spectrum synthesis, • *(ii) Bluetooth Low Energy (BLE)* for real-valued received signal strength indicator (RSSI) synthesis, • *(iii) 5G Cellular Network* for complex-valued channel state information (CSI) [38] synthesis.

### 5.1 RFID Spatial Spectrum Synthesis

TASK. Given a transmitter sending RF signals at location $(x_{tx}, y_{tx}, z_{tx})$, the goal is to synthesize the spatial spectrum received by the receiver (equipped with an antenna array). The spatial spectrum, represented as a $360 \times 90$ matrix, captures the signal power from all directions around the receiver, covering azimuth and elevation angles at a one-degree resolution. The elevation angle is limited to $90°$ as only the front hemisphere of the antenna array is considered [12].

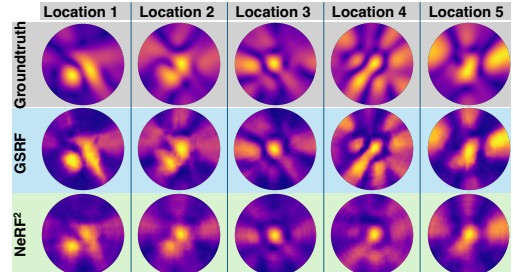

Figure 2: Visualization comparison of synthesized spatial spectrum at different positions.

DATASET. The publicly released RFID dataset from NeRF² [12], collected in real-world indoor environments, is employed. It contains 6,123 transmitter (RFID tag) locations and their corresponding spatial spectra, received by a receiver equipped with a $4 \times 4$ antenna array operating at the 915 MHz frequency band. The dataset is randomly split by default into 70% for training and 30% for testing.

METRICS. We employ the two metrics: • *(i) Mean Squared Error (MSE)↓*: This metric calculates the average of the squared differences in signal power between the synthesized spectrum and the ground truth for each entry. • *(ii) Peak Signal-to-Noise Ratio (PSNR, in dB)↑*: Treating the spatial spectrum as an image, PSNR measures structural similarity, with higher values indicating better quality.

BASELINES. We compare GSRF with NeRF² [12] and WRF-GS [34]. Other simulation-based or physics-unaware DL-based methods, such as MATLAB simulation [26], DCGAN [39], and VAE [30], perform worse on the same RFID dataset [12] compared to NeRF².

**Overall Performance.** To evaluate GSRF's performance in scenarios with insufficient data, we randomly select 220 instances from the training dataset instead of using the full training data. This creates a sparse dataset with a measurement density of $0.8$ measurements/ft³. Figure 2 presents the real-collected spatial spectra for four randomly selected transmitter positions (first row), alongside those generated by baseline models. Visually, the spectra synthesized by GSRF more closely match the ground truth compared to those by NeRF².

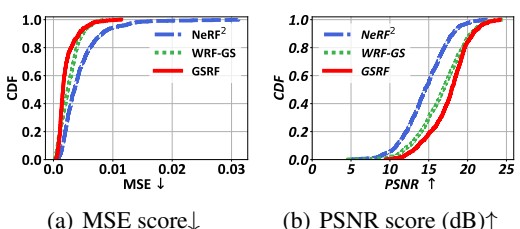

(a) MSE score↓      (b) PSNR score (dB)↑

Figure 3: Metric comparison for a sparse measurement density of $0.8$ measurements/ft³.

Figure 3 then shows the Cumulative Distribution Function (CDF) of the two metric scores on the testing data. GSRF achieves median improvements of 21.2% in PSNR and 56.4% in MSE over NeRF², while outperforming WRF-GS by 5.7% and 19.3%, respectively. This superiority stems from two advantages: First, our complex-valued Gaussian representation explicitly models phase interactions throughout the architecture, which is critical for RF signal propagation,

whereas WRF-GS relies on real-valued Gaussian primitives. Second, the Fourier-Legendre radiance basis in GSRF provides directional resolution beyond the spherical harmonics used in WRF-GS, enabling finer capture of diffraction and scattering effects. These innovations allow GSRF to learn more physically accurate scene representations even from sparse measurements.

**Training & Inference Efficiency.** Training time is measured by running each method on a computer equipped with GeForce RTX 3080Ti GPU. Inference time for each model is also recorded. Figure 4 illustrates that our method achieves convergence in 0.27 hours, which is 18.56× faster than NeRF$^2$ (5.01 hours) and 5.96× faster than WRF-GS (1.61 hours). For inference latency, as shown in Figure 5, our method synthesizes spatial spectra in 4.18 ms, yielding an 84.39×

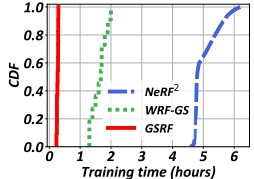 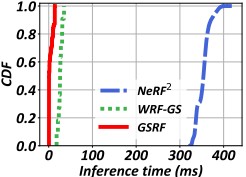

Figure 4: Training times for spectrum synthesis.

Figure 5: Test times for spectrum synthesis.

speedup over NeRF$^2$ (352.73 ms) and a 1.81× speedup over WRF-GS (7.58 ms). This acceleration is due to: *First,* our explicit Gaussian representation with FLE eliminates the need for MLP queries, which are required in both NeRF$^2$ and WRF-GS. Although WRF-GS also employs Gaussians, it still relies on a large MLP to query each Gaussian primitive's values using the Gaussian mean as input. *Second,* the hybrid CUDA-based ray tracer optimizes complex-valued operations through explicit gradient computation. These optimizations enable our method to support real-time applications, *e.g.*, sub-millisecond tracking in 5G networks.

**Measurement Density.** Figure 6 compares the MSE of GSRF (trained on the dataset with a density of 0.8 measurements/ft$^3$) to NeRF$^2$ (varying densities ranging from 0.8 to 15.5). The densities are obtained by random sampling from the original 70% training set. GSRF achieves a comparable MSE to NeRF$^2$ trained on the dataset with a density of 7.8. This indicates that GSRF requires 9.8× less training data to achieve comparable

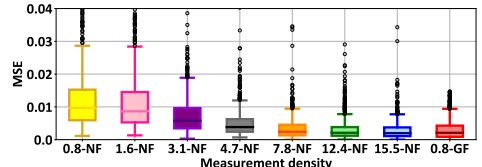

Figure 6: GSRF (GF) at 0.8 measurements/ft$^3$ and NeRF$^2$ (NF) across different densities.

spectrum synthesis quality to NeRF$^2$. The improvement arises from GSRF's 3D Gaussian-based scene representation, which focuses on object features rather than empty space, making it more efficient than NeRF$^2$'s voxel-based fields. More results for WRF-GS [34] are provided in Appendix F.4.

## 5.2 5G Complex-Valued CSI Synthesis

TASK. This task demonstrates GSRF's effectiveness in synthesising complex-valued signals. In 5G Orthogonal Frequency-Division Multiplexing (OFDM) modulation, downlink and uplink operate on different frequency bands [40]. Given uplink complex-valued CSI, the objective is to predict the downlink CSI. The rationale for this task lies in the shared physical propagation environment, which correlates uplink and downlink CSI [41]. Furthermore, uplink CSI can serve as a transmitter position indicator due to its uniqueness across different positions [12, 42].

DATASET. The public Argos dataset [43] is employed. It is collected in outdoor environments, where a base station with 104 antennas measures CSI from signals sent by clients. Each CSI measurement includes 52 subcarriers. Following prior works [12, 30, 41], the first 26 subcarriers are treated as the uplink channel, and the remaining $N = 26$ as the downlink channel. The dataset contains 100,000 measurements and is randomly split into 70% for training and 30% for testing.

METRICS. We adopt the Signal-to-Noise Ratio (SNR) [30] to quantify synthesized CSI quality:

$$\text{SNR} = -10 \log_{10} \left( \|S - \hat{S}\|_2^2 \cdot \|S\|_2^{-2} \right), \tag{17}$$

where $S, \hat{S} \in \mathbb{C}^N$ are the ground truth and synthesized CSI vectors, respectively.

BASELINES. We compare GSRF with NeRF$^2$ [12] and include two additional baselines:

• R2F2 [41]: Extracts the number of propagation paths and each path's parameters to estimate CSI.

• FIRE [30]: Uses the VAE [44] to predict the downlink CSI by learning the latent distribution.

**Overall Performance.** To demonstrate GSRF's efficiency, it is trained on only 30% of the raw training data portion, while the baselines are trained on the full training set. All methods are evaluated

on the same testing data. Since GSRF requires three-dimensional transmitter locations, we train an autoencoder [45] using 26 uplink subcarriers as input to reconstruct them. The autoencoder's hidden layer is set to three dimensions, representing the transmitter locations. Figure 7 illustrates a prediction example from GSRF, where the two curves (blue and red) nearly overlap, demonstrating its high prediction accuracy. Figure 8 quantifies the SNR of the four methods. GSRF achieves a mean SNR of 20.99 dB, outperforming R2F2 and FIRE. Additionally, GSRF achieves comparable CSI synthesis quality to NeRF$^2$ while using 3× less training data, highlighting its training efficiency.

It is worth noting that NeRF$^2$ also performs phase-aware modeling through an MLP that regresses amplitude and phase from voxel and transmitter coordinates. While this enables high-quality CSI synthesis, the volumetric ray-based querying of the MLP introduces significant computational cost during both training and inference. In contrast, GSRF integrates phase modeling directly into Gaussian primitives via Fourier–Legendre basis expansion, avoiding MLP regressors entirely.

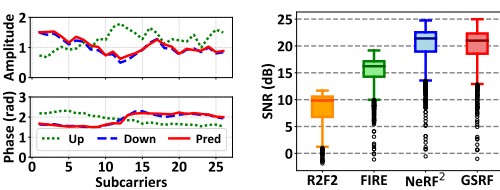

Figure 7: Channel amplitude and phase trace.

Figure 8: Channel CSI prediction SNR.

This explicit, complex-valued representation allows efficient gradient updates with lightweight CUDA operations, leading to faster convergence and inference without sacrificing accuracy. Thus, the comparable SNR to NeRF$^2$ does not diminish the contribution of GSRF, but instead underscores its ability to achieve phase-aware synthesis with substantially greater efficiency.

## 5.3 BLE Real-Valued RSSI Synthesis

TASK. This task verifies that GSRF supports single-antenna setups for capturing a single real-valued RSSI. Given a transmitter (BLE node) sending signals from location $(x_{tx}, y_{tx}, z_{tx})$, the goal is to synthesize the RSSI (in dBm) received by a receiver (BLE gateway with a single antenna). The measured RSSI represents the aggregate signal power from all directions [12]. Additionally, we conduct a fingerprint-based localization application to demonstrate GSRF's sensing advantages.

DATASET. The public BLE dataset [12], collected in an elderly nursing home, is employed. Twenty-one receivers operating at 2.4 GHz frequency band to capture RSSI. The dataset contains 6,000 transmitter positions, each paired with a 21-dimensional tuple of RSSI readings from the 21 receivers.

METRICS. *RSSI synthesis error* is the absolute difference between predictions and ground truth.

BASELINES. We compare GSRF with NeRF$^2$. Other empirical and DL methods, *e.g.*, MRI [46], are excluded because they perform worse than NeRF$^2$ on the same testing dataset [12].

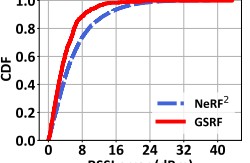
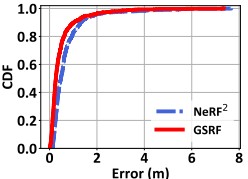

Figure 9: RSSI errors on the BLE dataset.

Figure 10: BLE-based localization error.

**Overall Performance.** To evaluate the performance of GSRF in scenarios with sufficient data, both models (GSRF and NeRF$^2$) are trained on the full training dataset. Figure 9 indicates that GSRF achieves an average RSSI error of 4.09 dBm, compared to NeRF$^2$'s 6.09 dBm. This represents a 32.79% improvement, highlighting GSRF's effectiveness fo single-antenna receivers. The performance gain stems from GSRF's flexible 3D Gaussian-based explicit scene representation, which efficiently utilizes training data by focusing on objects rather than large empty space and aligning with object geometry. Training and inference times are reported in Appendix F.2, Figures 11 and 12. These results demonstrate that GSRF achieves a 15.82-fold decrease in training time and a 78.98-fold reduction in inference time for RSSI synthesis. Additional results for WRF-GS [34] are in Appendix F.2.

**BLE-Based Localization.** In fingerprinting-based localization, the RSSI value from an unknown transmitter queries a fingerprint database containing pairs of transmitter positions and corresponding RSSI values. The K Nearest Neighbors (KNN) identifies the $K$ nearest matches and estimates the unknown transmitter position as the average of these $K$ positions [29]. We generate synthetic datasets using GSRF and NeRF$^2$ to build the fingerprint database for comparison. Figure 10 shows that GSRF outperforms NeRF$^2$ by 31.40% on average. This improvement in localization accuracy demonstrates that high-fidelity synthesized databases generated by GSRF enhance localization applications, eliminating the need for time-consuming and labor-intensive manually collected fingerprinting databases.

## 5.4 Ablation Study

We evaluate our design components using the RFID dataset introduced in Section 5.1. All versions are trained on the full training set, with results presented in Table 1.

**FLE-Based Radiance.** We employ FLE coeffi- cients to model the directional radiance of each Gaussian in GSRF, unlike the SH coefficients used in 3DGS for visible light rendering [8]. Ex-

Table 1: Effectiveness of components of GSRF.

| Metric | Radiance | Phase | Fourier loss | GSRF |
|--------|----------|-------|--------------|------|
| PSNR↑  | 20.51    | 20.89 | 21.30        | 22.64 |

perimental comparisons (first *vs.* last column) show that our FLE coefficients achieve better spectrum synthesis compared to SH coefficients. This improvement arises from FLE's ability to capture intricate RF signal interactions, *e.g.*, phase-dependent interference, while SH is more suitable for smooth optical functions. Both FLE and SH are implemented with a degree of $L = 3$, resulting in 16 coefficients each. Thus, GSRF enhances spatial spectrum synthesis by leveraging FLE's capability to handle complex-valued radiance fields.

**Phase Information.** Each Gaussian primitive represents radiance and transmittance as complex-valued attributes to capture RF signal propagation effects, such as constructive and destructive interference. Removing the phase channel while retaining only the amplitude results in a 8.37% reduction in PSNR compared to the full model with phase inclusion (second *vs.* last column). This performance drop underscores the importance of phase information in synthesizing RF signal data.

**Fourier Loss** $\mathcal{L}_{\textbf{Fourier}}$**.** We evaluate the impact of Fourier loss $\mathcal{L}_{\text{Fourier}}$ by comparing the model without this loss term (third column) against the full model (last column). Removing Fourier loss $\mathcal{L}_{\text{Fourier}}$ results in a 6.28% reduction in PSNR, indicating that frequency-domain alignment enhances the fidelity of the synthesized spatial spectra. Incorporating $\mathcal{L}_{\text{Fourier}}$ enables our model to better preserve frequency-domain properties, enhancing overall RF data synthesis quality.

## 6 Discussion

Despite its advancements, our method has two main limitations. It achieves efficient RF signal synthesis when training data is available for a specific scene but lacks spatial generality for zero-shot inference in unseen environments. It is also optimized for static settings: when the scene changes (*e.g.*, moving obstacles or structural modifications), retraining or fine-tuning is required, limiting temporal adaptation. To address these issues, we outline two complementary directions: improving spatial generality across environments and enabling temporal adaptability to dynamic scenes.

*For spatial generality,* future work will explore pre-training GSRF on large and diverse multi-scene RF datasets to learn transferable priors that capture common propagation patterns across environments. This could involve designing domain-general encoders that disentangle scene-invariant propagation features (*e.g.*, free-space loss, reflection/diffraction signatures) from scene-specific geometry, and leveraging domain-adaptation strategies to enable rapid adaptation to new environments with only a few samples. Another promising direction is hierarchical Gaussian representations, where global Gaussians encode universal priors while local Gaussians specialize to environment-specific details.

*For temporal adaptability,* we propose a deformable 3DGS extension that supports dynamic RF scene rendering: a shared set of complex-valued 3D Gaussians represents the baseline RF field, while a lightweight deformation module models time-varying changes without full retraining. A spatiotemporal encoder factors the 4D space–time volume $(x, y, z, t)$ into six compact 2D planes $(xy, yz, xz, xt, yt, zt)$, reducing parameter complexity from $R^4 \times C$ to $6R^2 \times C$ and preserving locality for efficient CUDA querying; spatial planes capture multipath effects (reflection/diffraction), and temporal planes capture motion-induced changes. A small multi-head decoder then predicts per-Gaussian deformations (position/rotation/scale), while complex-valued attributes (*e.g.*, radiance $\psi_k$, transmittance $\rho_k$) are preserved for phase-aware modeling. Together, these directions aim to make GSRF both broadly generalizable and responsive to real-world dynamics.

## 7 Conclusion

This paper introduces GSRF, a novel complex-valued 3DGS-based framework for efficient RF signal data synthesis. We customize 3D Gaussian primitives with complex-valued attributes and integrate an RF-specific CUDA-enabled ray tracing algorithm for efficient scene representation and received signal computation. Extensive experiments validate GSRF's efficiency, demonstrating significant improvements in training and inference speed while maintaining high-fidelity RF data synthesis.

## Acknowledgement

This research was funded in part by the Air Force Office of Scientific Research under awards # FA95502210193 and FA95502310559, and the DEVCOM Army Research Laboratory under award # W911NF-17-2-0196. Wan Du was partially supported by NSF Grant # 2239458, a UC Merced Fall 2023 Climate Action Seed Competition grant, and a UC Merced Spring 2023 Climate Action Seed Competition grant. Kang Yang was partially supported at UC Merced by a financial assistance award approved by the Economic Development Administration's Farms Food Future program. Sijie Ji's research is supported through a Schmidt Science Fellowship. Mani Srivastava was also partially supported by the Mukund Padmanabhan Term Chair at UCLA.

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

# A    Complex-Valued Ray Tracing Algorithm

This section derives Equation (14) for the received RF signal $S \in \mathbb{C}$ in GSRF, integrating wave propagation physics with Gaussian discretization.

## A.1    Continuous Wave Propagation Model

Consider a ray $\gamma(t) = \mathbf{r} + t\hat{\mathbf{v}}$ originating from the receiver at position $\mathbf{r}$ with direction $\hat{\mathbf{v}}$. The received signal is modeled as an integral along the ray path:

$$S = \int_0^\infty \psi(t)T(t)\,dt,$$

where:

- $\psi(t) \in \mathbb{C}$: Emitted signal at position $\gamma(t)$, defined by its amplitude $|\psi(t)|$ and phase $\angle\psi(t)$.
- $T(t) \in \mathbb{C}$: Cumulative transmittance from the receiver at $\gamma(0)$ to position $\gamma(t)$, given by:

$$T(t) = \exp\left(-\int_0^t \sigma(s)\,ds\right),$$

    where $\sigma(s) = \alpha(s) + j\beta(s)$ is the complex attenuation coefficient, with:
    - $\alpha(s) \geq 0$: Amplitude attenuation.
    - $\beta(s)$: Phase propagation.

## A.2    Discretization via 3D Gaussians

The scene is approximated using a set of 3D Gaussians $\{\zeta_k\}$, each specified by:

$$\zeta_k = (\mu_k, \Sigma_k, \psi_k, \sigma_k), \quad \sigma_k = \alpha_k + j\beta_k,$$

where:

- $\mu_k \in \mathbb{R}^3$: The mean position of the Gaussian.
- $\Sigma_k \in \mathbb{R}^{3\times3}$: The covariance matrix representing the spatial extent.
- $\psi_k \in \mathbb{C}$: The radiance, encoding amplitude and phase.
- $\sigma_k \in \mathbb{C}$: The complex attenuation coefficient, where:
    - $\alpha_k \geq 0$: Governs amplitude attenuation.
    - $\beta_k$: Determines phase propagation.

**Ray-Gaussian Intersection**

For each Gaussian $k$, we determine the intersection points of the ray $\gamma(t) = \mathbf{r} + t\hat{\mathbf{v}}$ with the Gaussian's ellipsoid, as detailed in Section 4.3. We solve:

$$(\gamma(t) - \mu_k)^\top \Sigma_k^{-1}(\gamma(t) - \mu_k) = 3,$$

which defines the ellipsoid boundary at the 3-sigma level. This yields the entry and exit times, $t_{\mathrm{in},k}$ and $t_{\mathrm{out},k}$, respectively. The path length through the Gaussian is then given by:

$$\Delta t_k = t_{\mathrm{out},k} - t_{\mathrm{in},k}.$$

**Midpoint Approximation**

The contribution of Gaussian $k$ is evaluated at the midpoint of the intersection segment:

$$t_{\mathrm{mid},k} = \frac{t_{\mathrm{in},k} + t_{\mathrm{out},k}}{2}, \quad \mathbf{x}_{\mathrm{mid},k} = \gamma(t_{\mathrm{mid},k}).$$

This approximation introduces an error of order $O(\Delta t_k^3)$, which is negligible for small $\Delta t_k$, as is common in dense Gaussian representations.

**Transmittance Derivation**

Transmittance $\rho_m$ for Gaussian $m$ accounts for attenuation and phase shift over the path length $\Delta t_m$:

$$\rho_m = \exp\left(-\int_{t_{\mathrm{in},m}}^{t_{\mathrm{out},m}} \sigma_m\, dt\right) = e^{-\sigma_m \Delta t_m} = \underbrace{e^{-\alpha_m \Delta t_m}}_{|\rho_m|} \cdot \underbrace{e^{-j\beta_m \Delta t_m}}_{e^{j\angle \rho_m}},$$

where:

- $|\rho_m| = e^{-\alpha_m \Delta t_m} \leq 1$: Amplitude transmission factor, since $\alpha_m \geq 0$.
- $\angle \rho_m = -\beta_m \Delta t_m$: Phase shift of the transmitted signal.

### A.3  Discretized Rendering Equation

The Gaussians are sorted by their midpoint times $t_{\mathrm{mid},k}$ (in increasing distance from the receiver) to ensure correct depth ordering. The discretized received signal is expressed as:

$$S = \sum_{k=1}^{K_{\mathrm{intr}}} \mathcal{G}_k(\mathbf{x}_{\mathrm{mid},k}) \cdot \psi_k \cdot \prod_{m=1}^{k-1} \rho_m,$$

where:

- $\mathcal{G}_k(\mathbf{x}_{\mathrm{mid},k})$: The Gaussian density at the midpoint, given by:

$$\mathcal{G}_k(\mathbf{x}) = \frac{1}{(2\pi)^{3/2}|\Sigma_k|^{1/2}} \exp\left(-\frac{1}{2}(\mathbf{x}-\mu_k)^\top \Sigma_k^{-1}(\mathbf{x}-\mu_k)\right).$$

- $\psi_k = |\psi_k|\, e^{j\angle \psi_k}$: The emitted radiance of Gaussian $k$.
- $\prod_{m=1}^{k-1} \rho_m$: This term represents the cumulative transmittance from the receiver through all preceding Gaussians up to $k-1$.

## B  Gaussian Primitive Optimization

The following two strategies are employed to update the number of Gaussians and their attributes.

**Attribute Updating** Each 3D Gaussian stores its own attributes and updates them using SGD [47]:

$$w^{(j+1)} = w^{(j)} - \eta_w \cdot \nabla_w \mathcal{L}\left(w^{(j)}\right) \tag{18}$$

where $w$ represents any attribute of a Gaussian, each with its own learning rate $\eta_w$. The term $\nabla_w \mathcal{L}\left(w^{(j)}\right)$ denotes the gradient of the loss function $\mathcal{L}$ with respect to $w$ at iteration $j$. For radiance $\psi$, the updated parameters are the FLE coefficients.

The covariance matrix $\Sigma$ is physically meaningful only when positive semi-definite [48], but the update equation above does not guarantee this property. To address this, we adopt the solution proposed in [8], which represents $\Sigma = RSS^T R^T$, where $R$ is a rotation matrix and $S$ is a scaling matrix. Updates are applied independently to $R$ and $S$, ensuring that $\Sigma$ remains positive semi-definite.

**Number of Gaussian Updating.** The initial number of Gaussians is set by cube-based initialization. However, this number is suboptimal, as some areas require more Gaussians (*e.g.*, object regions), while others need fewer (*e.g.*, free space) to model RF signal propagation effectively. We observe that such cases lead to large gradients for the Gaussian's mean $\mu$, as the existing 3D Gaussians do not adequately capture the area's effect on RF signal propagation. The mean $\mu$ exhibits larger gradients than other attributes because it represents the position with the highest probability, making it crucial for modeling RF signal behavior.

To this end, we employ a gradient-threshold-based strategy: *First*, every $N_\mu$ iterations, we compute the average gradient of the mean $\mu$ for all Gaussians and select those with a mean gradient exceeding a threshold $\epsilon_\mu$. *Second*, we determine the radius of each selected Gaussian, approximated as the average of the diagonal values of its covariance matrix. A radius threshold $\epsilon_r$ classifies them as small or large Gaussians. *Third*, small Gaussians are cloned by duplicating them and shifting the copies in the direction of the gradient. Large Gaussians are split into two new Gaussians, reducing their scaling

---

**Algorithm 1:** Forward CUDA Kernel for Ray Tracing Algorithm

---

**Input:** $w, h$: numbers of rays in azimuth and elevation
**Input:** $M, C$: means & covariances of all Gaussians
**Input:** $E, A$: radiances & transmittances of all Gaussians
**Input:** $L$: positions of receiver and transmitter
**Output:** $O$: received signals for all rays

---

1 **Function** RayTracing(*w, h, M, C, E, A, L*):
2     $M', C' \leftarrow$ sphericalGaussian($M, C, L$)
3     Grids $\leftarrow$ buildGrid($w, h$)
4     Idx, Kys $\leftarrow$ sphericalSplatting($M'$, Grids)
5     Ranges $\leftarrow$ computeGridRange(Idx, Kys)
6     $O \leftarrow 0$
7     **forall** *grid G in Grids* **do**
8        **forall** *ray i in G* **do**
9           ra $\leftarrow$ getGridRange(Ranges, $g$)
10           $O[i] \leftarrow$ Blend($i$, Idx, ra Kys, $M'$, $C'$, $E$, $A$)
11        **end**
12     **end**
13     **return** $O$

---

matrix $R$ by a factor of $\phi$ and initializing their positions by sampling from the original Gaussian's PDF. Additionally, every $N_\rho$ iterations, we remove Gaussians with attenuation $\rho$ below a threshold $\epsilon_\rho$, as they minimally impact signal propagation, *e.g.*, in free space. A single 3D Gaussian distribution can represent a large free space.

## C   CUDA Kernel

We develop two CUDA kernels for the forward and backward computations for RF signal renderer.

**Forward Kernel.** Algorithm 1 outlines the forward kernel. The inputs include the number of rays in azimuth and elevation, the means, covariance matrices, radiances, and attenuations of all 3D Gaussians, as well as the positions of the receiver and transmitter. The output is the received signal computed for all $N_{\text{az}} \times N_{\text{el}}$ rays.

Specifically, Line 2 projects the 3D Gaussians onto the 2D RF plane. Line 3 partitions all rays into multiple grids, each containing $N_{\text{rays}}$ rays in the azimuth and elevation directions, to accelerate processing. Line 4 applies the splatting process to identify which Gaussians influence each grid. Line 5 records the sorted Gaussians within each grid. Finally, Lines 7–12 compute the received signal for each ray in parallel using the complex-valued blending algorithm.

**Backward Kernel.** Since the Forward Kernel is invoked for ray tracing forward computation, PyTorch cannot automatically compute the corresponding computation graph gradients. After computing the received signal $S$ and the loss $\mathcal{L}$, PyTorch calculates the gradient $\frac{\partial \mathcal{L}}{\partial S}$, which is then passed to the Backward Kernel. This kernel reverses the computations of the Forward Kernel to compute the gradients for each Gaussian attribute. We explicitly derive the gradients for all Gaussian attributes for gradient-based attribute learning, with detailed computations provided in Appendix D.

## D   Gradient Computation

After computing the received signal $S$ and the loss $\mathcal{L}$, PyTorch calculates the gradient $\frac{\partial \mathcal{L}}{\partial S}$. We apply the chain rule [49] to compute the derivatives for each Gaussian in the Backward Kernel:

$$S = \sum_{k=1}^{K_{\text{intr}}} p_{\text{intr},k} \cdot |\psi_k| \, e^{j\angle\psi_k} \cdot \prod_{m=1}^{k-1} \rho_m,$$

where:

- $p_{\text{intr},k} = \mathcal{G}_k\left(x_{\text{rep},k}; \mu_k, \Sigma_k\right)$: The Gaussian weight.
- $\psi_k = |\psi_k| \, e^{j\angle\psi_k}$: The radiance of the $k$-th Gaussian.
- $\rho_m = |\rho_m| \, e^{j\angle\rho_m}$: The complex-valued transmittance.

### D.1 Gradient for k-th Gaussian Radiance

**Compute $\frac{\partial \mathcal{L}}{\partial |\psi_k|}$:**

$$\frac{\partial \mathcal{L}}{\partial |\psi_k|} = \frac{\partial \mathcal{L}}{\partial S} \cdot p_{\text{intr},k} \cdot e^{j \angle \psi_k} \cdot \prod_{m=1}^{k-1} \rho_m$$

**Compute $\frac{\partial \mathcal{L}}{\partial \angle \psi_k}$:**

$$\frac{\partial \mathcal{L}}{\partial \angle \psi_k} = \frac{\partial \mathcal{L}}{\partial S} \cdot j p_{\text{intr},k} \cdot |\psi_k| e^{j \angle \psi_k} \cdot \prod_{m=1}^{k-1} \rho_m$$

### D.2 Gradient for k-th Gaussian Transmittance

The transmittance $\rho_k$ influences all emissions from subsequent Gaussians $j = k+1, \ldots, K_{\text{intr}}$. The cumulative transmittance for Gaussian $j$'s radiance is defined as:

$$P_j = \prod_{m=1}^{j-1} \rho_m$$

**Compute $\frac{\partial \mathcal{L}}{\partial |\rho_k|}$:**

Partial derivative of $P_j$ with respect to $|\rho_k|$:

$$\frac{\partial P_j}{\partial |\rho_k|} = \begin{cases} 0 & \text{if } j \leq k \\ \frac{P_j}{|\rho_k|} & \text{if } j > k \end{cases}$$

Differentiating $S$ with respect to $|\rho_k|$:

$$\frac{\partial S}{\partial |\rho_k|} = \sum_{j=k+1}^{K_{\text{intr}}} p_{\text{intr},j} \cdot |\psi_j| e^{j \angle \psi_j} \cdot \frac{P_j}{|\rho_k|}$$

Using the chain rule:

$$\frac{\partial \mathcal{L}}{\partial |\rho_k|} = \frac{\partial \mathcal{L}}{\partial S} \cdot \sum_{j=k+1}^{K_{\text{intr}}} p_{\text{intr},j} \cdot |\psi_j| e^{j \angle \psi_j} \cdot \frac{P_j}{|\rho_k|}$$

**Compute $\frac{\partial \mathcal{L}}{\partial \angle \rho_k}$:**

The partial derivative of $P_j$ with respect to $\angle \rho_k$ is expressed as:

$$\frac{\partial P_j}{\partial \angle \rho_k} = \begin{cases} 0 & \text{if } j \leq k \\ j P_j & \text{if } j > k \end{cases}$$

Differentiating the loss with respect to the phase of $\rho_k$:

$$\frac{\partial \mathcal{L}}{\partial \angle \rho_k} = \frac{\partial \mathcal{L}}{\partial S} \cdot \sum_{j=k+1}^{K_{\text{intr}}} j \, p_{\text{intr},j} \cdot |\psi_j| e^{j \angle \psi_j} \cdot P_j$$

### D.3 Gradients for k-th Gaussian Mean and Covariance Matrix

For brevity, we omit the subscript $k$ in $p_{\text{intr}}$, $\mu$, and $\Sigma$, where it implicitly denotes the $k$-th Gaussian. Before computing the derivatives for the mean $\mu$ and covariance matrix $\Sigma$, we first calculate $\frac{\partial \mathcal{L}}{\partial p_{\text{intr}}}$:

$$\frac{\partial \mathcal{L}}{\partial p_{\text{intr}}} = \frac{\partial \mathcal{L}}{\partial S} \cdot |\psi| e^{j \angle \psi} \cdot \prod_{m=1}^{k-1} \rho_m$$

To compute the gradients for the mean $\mu$ and the covariance matrix $\Sigma$, we first examine the forward computation of the probability $p_{\text{intr}}$.

### D.3.1 Defining 3-Sigma Ellipsoid

$$(\mathbf{x} - \mu)^\top \Sigma^{-1}(\mathbf{x} - \mu) = 9$$

where:

- $\mathbf{x}$: A point in 3D space.
- $\mu$: The mean (center of the ellipsoid).
- $\Sigma$: The covariance matrix defining the ellipsoid's shape and orientation.

### D.3.2 Ray Parameterization

The parametric equation of a ray, as given in Equation (10), is:

$$\mathbf{x}(d) = \mathbf{l}_{\text{rx}} + d\hat{\mathbf{v}}, \quad d \geq r_{\text{rx}}$$

where:

- $\mathbf{l}_{\text{rx}}$: Receiver position.
- $\hat{\mathbf{v}}$: Unit direction vector of the ray.
- $d$: Distance parameter along the ray.
- $r_{\text{rx}}$: Radius of the Ray Emitting Spherical Surface (RESS), defining the starting point for each ray.

Substitute $\mathbf{x}(d)$ into the ellipsoid equation:

$$(\mathbf{l}_{\text{rx}} + d\hat{\mathbf{v}} - \mu)^\top \Sigma^{-1}(\mathbf{l}_{\text{rx}} + d\hat{\mathbf{v}} - \mu) = 9$$

Expanding the dot product:

$$(\mathbf{l}_{\text{rx}} - \mu)^\top \Sigma^{-1}(\mathbf{l}_{\text{rx}} - \mu) + 2d\hat{\mathbf{v}}^\top \Sigma^{-1}(\mathbf{l}_{\text{rx}} - \mu) + d^2\hat{\mathbf{v}}^\top \Sigma^{-1}\hat{\mathbf{v}} = 9$$

### D.3.3 Solving the Quadratic Equation

Rearrange the equation:
$$Ad^2 + 2Bd + (C - 9) = 0$$

where:
$$A = \hat{\mathbf{v}}^\top \Sigma^{-1}\hat{\mathbf{v}}, \quad B = \hat{\mathbf{v}}^\top \Sigma^{-1}(\mathbf{l}_{\text{rx}} - \mu), \quad C = (\mathbf{l}_{\text{rx}} - \mu)^\top \Sigma^{-1}(\mathbf{l}_{\text{rx}} - \mu)$$

Using the quadratic formula:

$$d_{1,2} = \frac{-B \pm \sqrt{B^2 - A(C - 9)}}{A}$$

The two intersection points along the ray are:

$$\mathbf{x}_1 = \mathbf{l}_{\text{rx}} + d_1\hat{\mathbf{v}}, \quad \mathbf{x}_2 = \mathbf{l}_{\text{rx}} + d_2\hat{\mathbf{v}}$$

The valid intersection is the one where $d \geq r_{\text{rx}}$. If no valid $d$ exists, the ray does not intersect the ellipsoid:

$$\mathbf{x}_{\text{intr}} = \frac{\mathbf{x}_1 + \mathbf{x}_2}{2} = \mathbf{l}_{\text{rx}} + \frac{d_1 + d_2}{2}\hat{\mathbf{v}}$$

The probability of intersection is then computed as:

$$p_{\text{intr}} = \frac{1}{(2\pi)^{3/2}|\Sigma|^{1/2}} \exp\left(-\frac{1}{2}(\mathbf{x}_{\text{intr}} - \mu)^\top \Sigma^{-1}(\mathbf{x}_{\text{intr}} - \mu)\right)$$

### D.3.4 Backward Computation for Mean

**Compute $\frac{\partial p_{\text{intr}}}{\partial \mathbf{x}_{\text{intr}}}$ and $\frac{\partial p_{\text{intr}}}{\partial \mu}\Big|_{\text{direct}}$:**

Since $p_{\text{intr}}$ depends on both $\mathbf{x}_{\text{intr}}$ and $\mu$, we compute the gradients as follows:

$$\frac{\partial p_{\text{intr}}}{\partial \mathbf{x}_{\text{intr}}} = -p_{\text{intr}} \cdot \Sigma^{-1}(\mathbf{x}_{\text{intr}} - \mu)$$

$$\frac{\partial p_{\text{intr}}}{\partial \mu}\Big|_{\text{direct}} = p_{\text{intr}} \cdot \Sigma^{-1}(\mathbf{x}_{\text{intr}} - \mu)$$

**Compute $\frac{\partial \mathbf{x}_{\text{intr}}}{\partial \mu}$:**

$$\frac{\partial \mathbf{x}_{\text{intr}}}{\partial \mu} = \frac{1}{2}\left(\frac{\partial d_1}{\partial \mu} + \frac{\partial d_2}{\partial \mu}\right)\hat{\mathbf{v}}$$

**Compute $\frac{\partial d_1}{\partial \mu}$ and $\frac{\partial d_2}{\partial \mu}$:**

The partial derivatives of $d_1$ and $d_2$ with respect to the mean $\mu$ are given by:

$$\frac{\partial d_{1,2}}{\partial \mu} = \frac{-\frac{\partial B}{\partial \mu} \pm \frac{1}{2}(B^2 - A(C-9))^{-1/2}\left(2B\frac{\partial B}{\partial \mu} - A\frac{\partial C}{\partial \mu}\right)}{A}$$

where:

$$\frac{\partial B}{\partial \mu} = -\Sigma^{-1}\hat{\mathbf{v}}, \quad \frac{\partial C}{\partial \mu} = -2\Sigma^{-1}(\mathbf{l}_{\text{rx}} - \mu)$$

**Compute $\frac{\partial p_{\text{intr}}}{\partial \mu}$:**

The gradient of $p_{\text{intr}}$ with respect to the mean $\mu$ is computed using the chain rule as follows:

$$\frac{\partial p_{\text{intr}}}{\partial \mu} = \frac{\partial p_{\text{intr}}}{\partial \mathbf{x}_{\text{intr}}} \cdot \frac{\partial \mathbf{x}_{\text{intr}}}{\partial \mu} + \frac{\partial p_{\text{intr}}}{\partial \mu}\Big|_{\text{direct}}$$

**Compute $\frac{\partial \mathcal{L}}{\partial \mu}$:**

The gradient of the loss $\mathcal{L}$ with respect to the Gaussian mean $\mu$ is computed as:

$$\frac{\partial \mathcal{L}}{\partial \mu} = \frac{\partial \mathcal{L}}{\partial p_{\text{intr}}} \cdot \frac{\partial p_{\text{intr}}}{\partial \mu}$$

The gradient for the covariance matrix $\frac{\partial \mathcal{L}}{\partial \Sigma}$ follows similar steps as the mean gradient computation, so we omit it for brevity.

## E  Implementation

**Training.** Table 2 presents the hyperparameter settings, defined in Section 4. These values are determined through extensive empirical studies. The attributes of all 3D Gaussians are updated using SGD [47]. The learning rates are set as follows: $\eta_\rho = 0.01$ for attenuation, $\eta_\psi = 0.0025$ for emission, $\eta_S = 0.01$ for the scaling matrix, and $\eta_R = 0.005$ for the rotation matrix. The learning rate for the mean, $\eta_\mu$, starts at 0.00016 and de-

Table 2: Hyperparameter settings.

| Symbol | Meaning | Value |
|---|---|---|
| $\epsilon_\mu$ | Threshold for mean gradient | 0.0002 |
| $\epsilon_r$ | Threshold for radius | 10.0 |
| $\epsilon_\rho$ | Threshold for attenuation | 0.004 |
| $N_\mu$ | Gradient check frequency | 100 |
| $N_\rho$ | Attenuation check frequency | 100 |
| $r_{\text{rx}}$ | Radius of the RESS | 1.0 |
| $\phi$ | Scaling matrix reduction factor | 1.6 |

creases exponentially to $1.6 \times 10^{-6}$ over 30,000 iterations. For Gaussian count optimization, the number of Gaussians is optimized only during the first half of the total iterations. After that, only the attributes of the Gaussians are updated.

**CUDA Kernel.** Each grid contains 16 rays in both azimuth and elevation angles, totaling $N_{\text{rays}} = 16 \times 16$ rays per grid. Gaussians intersecting each grid are sorted using the CUDA built-in `cub::DeviceRadixSort` API [50]. Each splatting instance (a Gaussian intersecting a grid) is assigned a 64-bit key: the lower 32 bits store the distance to the receiver, while the upper 32 bits encode the grid index. This structure enables efficient parallel sorting of all splats by distance with a single invocation of the `cub::DeviceRadixSort` API.

To integrate PyTorch with CUDA execution, we implement a custom PyTorch extension using C++ and CUDA, enabling efficient GPU-accelerated computations. The forward and backward computations are encapsulated within a subclass of `torch.autograd.Function`, ensuring seamless differentiation and gradient propagation within PyTorch's computational graph. The Python interface, implemented via PyTorch's C++ API, facilitates interaction between PyTorch tensors and CUDA kernels, handling memory layout conversions and efficient CPU-GPU data transfers.

# F  Experimental Results

## F.1  Scenario Overview

This paper considers a scenario where a receiver is fixed at a position (*e.g.*, 5G base station or LoRa gateway), while a transmitter (*e.g.*, smartphone or LoRa node) can be at any location in 3D space. Given a dataset of some transmitter locations and their corresponding received signals, the goal is to predict the received signal from a transmitter at a new position.

Alternatively, the roles can be reversed: the transmitter is fixed (*e.g.*, WiFi router), while the receiver is placed at different locations (*e.g.*, smartphone). According to reciprocity between the transmitter and receiver [30], these two scenarios are essentially equivalent. Consequently, this work focuses solely on the first scenario.

## F.2  BLE RSSI Synthesis

**Training and Inference Time.** Both methods are trained for 100,000 iterations. Training time is measured by running each method 10 times on a computer equipped with an NVIDIA GeForce RTX 3080 Ti. Inference time for each model is also recorded. Figure 11 shows that GSRF reduces training time from 2.69 hours with NeRF² to 0.17 hours, achieving a 15.82-fold decrease. Similarly, Figure 12 illustrates that GSRF

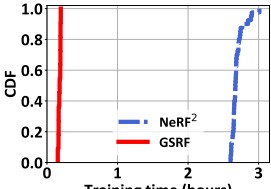
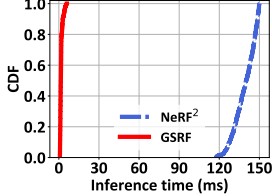

Figure 11: Training times for RSSI synthesis.

Figure 12: Inference times for RSSI synthesis.

reduces inference time from 139.01 ms with NeRF² to 1.76 ms, a 78.98-fold reduction. These short inference times enable GSRF to support real-time applications [30].

**BLE RSSI Prediction and Localization.** We extend the evaluation to include WRF-GS [34], trained on the full dataset, for both RSSI prediction and localization tasks. GSRF reduces RSSI error by 3.92% compared to WRF-GS, demonstrating the benefit of its unified RF modeling. Localization errors remain similar across models, reflecting the inherent resilience of the KNN baseline: by selecting the $k$ nearest neighbors in the RSSI fingerprints and averaging their positions, KNN effectively acts as a low-pass filter that mitigates synthesis noise.

Table 3: BLE RSSI prediction error and localization error across models.

|  | NeRF² | GSRF | WRF-GS |
|---|---|---|---|
| RSSI error (dBm) | 6.091±5.427 | 4.094±3.908 | 4.261±3.943 |
| Localization error (m) | 0.699±0.804 | 0.479±0.692 | 0.481±0.685 |

## F.3  Parameter Study

We further investigate the effect of several key hyperparameters in GSRF, including the Fourier–Legendre Expansion (FLE) degree, the angular resolution, and the number of Gaussians. These studies provide insights into the trade-offs between accuracy, efficiency, and stability, and guide the recommended default settings.

**Fourier–Legendre Expansion (FLE) Degree.** The FLE basis degree $L$ controls the expressiveness of GSRF 's complex-valued 3D Gaussians in modeling phase-aware RF propagation effects such as interference and diffraction. Low degrees ($L = 1$–$2$) capture only coarse angular components, which leads to underfitting. Moderate degrees ($L = 3$–$4$) capture essential variations efficiently, while high degrees ($L \geq 5$) risk overfitting and add computation. We find that $L = 3$ provides the best balance between accuracy and efficiency. Experimental results are summarized in Table 4.

Table 4: Effect of FLE degree $L$ on PSNR, training time, and inference speed.

| Degree $L$ | PSNR (dB) | Training time (min) | Inference time (ms) |
|---|---|---|---|
| 1 | 16.49 | 13.84 | 2.96 |
| 2 | 17.73 | 15.09 | 3.38 |
| 3 | 18.67 | 16.21 | 4.18 |
| 4 | 18.78 | 19.26 | 6.27 |
| 5 | 18.21 | 24.72 | 8.62 |

**Angular Resolution.** The angular resolution in GSRF is not a rigid hyperparameter but is instead governed by the antenna configuration and the measurement setup. Its role is to balance fidelity, coverage, and computational efficiency in modeling RF propagation.

For *multi-antenna arrays*, the effective angular resolution follows the spatial sampling theorem and scales with the number of array elements. For example, in our RFID dataset, a $4 \times 4$ uniform rectangular array supports $\sim 1°$ resolution using classical algorithms such as MUSIC (Multiple Signal Classification). We align the resolution with the measurement data: since the RFID dataset was collected at $1°$ intervals over azimuth and elevation, we preserve this resolution to avoid interpolation artifacts. If the measurement data had coarser sampling (*e.g.*, $2°$ intervals due to a smaller array), GSRF could operate at that resolution without modification, since both the orthographic splatting process and the loss functions are resolution-agnostic and work with arbitrary ray grids.

For *single-antenna configurations*, the received signal is inherently scalar, with no native angular resolution per antenna theory. In such cases (*e.g.*, RSSI synthesis), we discretize the spherical rendering at $1°$ to ensure dense coverage of propagation paths. This choice is flexible: finer bins (*e.g.*, $0.5°$) increase the ray count without proportional fidelity gains at centimeter wavelengths, while coarser bins (*e.g.*, $5°$) reduce runtime but risk missing important multipath effects. Our $1°$ setting thus represents a practical trade-off, and it is consistent with conventions in RF Computer-Aided Design (CAD) simulation tools such as Wireless InSite and the MATLAB Ray Tracing toolbox.

*Experimental Validation.* Table 5 reports the effect of angular resolution in the single-antenna RSSI synthesis task (Section 5.3). At $1°$, with $360 \times 90 = 32{,}400$ rays, GSRF achieves the lowest RSSI error due to dense angular sampling. As the resolution coarsens, accuracy degrades: $2°$ resolution produces slightly higher error (with $180 \times 45 = 8{,}100$ rays), and $5°$ resolution degrades further ($72 \times 18 = 1{,}296$ rays). Training and inference times scale proportionally, demonstrating a tunable trade-off: high-resolution settings suit precision-critical applications, whereas coarser settings may be preferable when computational efficiency is paramount.

Table 5: Effect of angular resolution on RSSI synthesis.

| Resolution | RSSI error (dBm) | Training time (min) | Inference time (ms) |
|---|---|---|---|
| $1°$ | 4.094 | 10.23 | 1.76 |
| $2°$ | 4.493 | 8.56 | 0.94 |
| $5°$ | 6.518 | 4.92 | 0.17 |

**Number of Gaussians.** The number of Gaussians is not a manually fixed hyperparameter. As detailed in Appendix B and following the original 3DGS [8], it is dynamically optimized through densification and pruning during training. This process automatically adds Gaussians in under-reconstructed regions and prunes redundant ones, ensuring the model adaptively balances representation capacity and efficiency without manual tuning.

**Cube-Based Initialization.** The cube-based initialization in GSRF refers to a uniform strategy for placing Gaussian primitives at the start of training. This choice is motivated by the need for comprehensive coverage of the 3D scene volume, which accelerates convergence. By distributing Gaussians uniformly across the bounding box that encloses the transmitter, receiver, and environment, the model begins with a balanced representation of potential RF propagation paths. This avoids

early coverage gaps that could arise from sparse or random initialization, enabling the subsequent densification and pruning process to refine the representation more effectively. While random initialization can achieve similar fidelity after sufficient optimization, it typically requires longer training time (0.59 hours for random initialization *vs.* 0.27 hours for uniform initialization on the RFID spatial spectrum synthesis task).

### F.4 Measurement Density

We extend the measurement density in Section 5.1 analysis to include WRF-GS [34]. When trained with 0.8 measurements/ft$^3$, WRF-GS and GSRF achieve MSEs of $0.002659 \pm 0.003560$ and $0.002147 \pm 0.003343$, respectively, both comparable to NeRF$^2$'s $0.002405 \pm 0.003623$ despite the latter being trained with a substantially higher density of 7.8 measurements/ft$^3$. This advantage arises from the explicit 3DGS representation adopted by WRF-GS and GSRF, where Gaussian primitives provide greater representational power and flexibility than NeRF-based volumetric sampling, thereby improving efficiency under sparse measurement conditions.

### F.5 Practical Benefits

**RFID.** An angular artificial neural network (AANN) identifies the Angle of Arrival (AoA) of line-of-sight path from received spatial spectra, enabling spectrum-based localization [51]. The AANN is trained on pairs of spectra and their corresponding AoA labels. Both GSRF and NeRF$^2$ can synthesize spectra for AANN training. Compared to NeRF$^2$, adopting GSRF can significantly reduce real-world resource consumption. For example, in a conference room ($26.2\,ft \times 16.4\,ft \times 9.8\,ft$) [52] with a measurement time of one minute per measurement, reducing the measurement density from 7.8 to 0.8 measurements/ft$^3$ saves approximately 200 hours of data collection time. Additionally, GSRF reduces computing time by 5.71 hours, including 4.74 hours for training and 0.97 hours for inference, both of which greatly save computational resources.

**BLE.** Similar to the previous field study, GSRF eliminates the need for site surveys, significantly reducing data collection time. Its fast training GPU-hours and low inference latency save server computation resources, accelerating the construction of the fingerprint database.

**5G.** The current method for obtaining downlink CSI requires client feedback, causing significant transmission overhead [30]. GSRF eliminates this overhead. Furthermore, GSRF's low inference latency makes it suitable for 5G networks. In contrast, NeRF$^2$'s inference latency of over 300 ms exceeds the coherence time in dynamic scenarios [30], making it impractical for 5G applications.

## G Design Discussion

**Why Fourier–Legendre Basis Instead of Spherical Harmonics.** We adopt Fourier–Legendre Expansion (FLE) over the more common Spherical Harmonics (SH) to model directional radiance in GSRF. This choice is motivated by the fundamental differences between RF propagation and visible light rendering, as well as the mathematical properties of the two bases. RF signals at centimeter-scale wavelengths exhibit pronounced phase-dependent interference and diffraction, which SH is ill-suited to capture efficiently.

*Limitations of SH in RF.* SH provides an orthogonal basis on the sphere and is widely used in 3DGS for representing smooth, low-frequency view-dependent effects in the visible domain (*e.g.*, shading, reflections). However, SH suffers from two limitations in RF applications. First, its low-frequency bias makes it converge slowly for oscillatory patterns: centimeter-wavelength RF fields often exhibit sharp constructive and destructive interference, leading to high-frequency angular variations that SH requires high degrees to approximate. Second, standard SH relies on real coefficients and is therefore phase-insensitive, making it poorly suited for modeling complex-valued RF fields where phase differences govern interference outcomes.

*Advantages of FLE.* FLE combines Fourier series (for azimuthal periodicity) with Legendre polynomials (for elevation dependency), and is better aligned with the properties of RF propagation. Its Fourier component naturally captures periodic phase shifts and oscillatory interference patterns, while Legendre polynomials provide orthogonal support over elevation. Unlike SH, FLE employs complex coefficients, allowing direct encoding of both amplitude and phase, which is essential for accurate RF modeling. Moreover, the polar–azimuthal decomposition of FLE matches the geometry of antenna measurements over spherical regions, providing better locality and efficiency for multipath effects compared to the global harmonics of SH.

Overall, FLE offers a more compact, phase-aware, and physically aligned basis for directional RF radiance, enabling GSRF to efficiently capture the interference-rich characteristics of RF propagation.

**Role of SSIM Loss.** While the blob-like primitives in 3DGS representations can produce smoother outputs, the Structural Similarity Index (SSIM) remains valuable as a complementary loss. Unlike L1, which emphasizes pixel-wise accuracy, SSIM emphasizes structural and perceptual similarity, making it particularly suitable for RF synthesis tasks where outputs such as spatial spectra are image-like data (*e.g.*, directional signal power). This follows standard practice in prior 3DGS methods (*e.g.*, 3DGS [8] and WRF-GS [34]), which incorporate SSIM to enhance perceptual quality. Our ablation confirms its effectiveness: removing SSIM and relying solely on L1 and Fourier loss reduces RFID spatial spectrum synthesis PSNR by 0.73 dB (from 22.64 to 21.91 dB), indicating that SSIM helps refine structural details in the synthesized spectra.

**Equirectangular Projection *vs*. Cube-Map.** We adopt equirectangular projection for representing RF spatial spectra, despite its known polar stretching, due to its simplicity and compatibility with azimuth–elevation parameterizations commonly used in antenna array data. This choice enables uniform angular sampling without additional remapping and aligns directly with the latitude–longitude grids of collected datasets. The distortion near high elevations has limited practical impact in RF scenarios, since paths above $60°$ typically correspond to ceilings, floors, or skyward directions where signals are heavily attenuated or yield few useful multipath components. Although cube-maps could mitigate polar distortion, they introduce seam artifacts and gradient discontinuities across faces, destabilizing backpropagation in a differentiable rendering pipeline. Moreover, cube-maps exhibit non-uniform sampling density across faces, which conflicts with the uniform angular resolution of RF spatial spectrum measurements. Thus, equirectangular projection provides a more stable and dataset-aligned choice for our framework.

**Multipath Effects.** GSRF is designed to capture multipath effects by representing the RF scene as a collection of complex-valued 3D Gaussians, where each Gaussian acts as a primitive that approximates a propagation path or interaction point. Multipath propagation introduces amplitude attenuation and phase shifts across different paths, which are modeled through complex-valued radiance and transmittance attributes encoded via the Fourier–Legendre basis. Ray tracing plays a critical role: rays are emitted from the receiver across a spherical surface, and the contributions of intersecting Gaussians are aggregated. Transmittance encodes path-length–dependent phase shifts and attenuation, and the summation of complex contributions enables both constructive and destructive interference. This effectively discretizes the continuous wave propagation integral, ensuring that path-specific interactions are preserved. Without ray tracing, aggregation would reduce to simple amplitude blending, which is insufficient for centimeter-scale RF modeling.

Empirical validation is performed using real-collected datasets, where explicit ground truth for multipath components is not directly observable due to measurement aggregation. Instead, multipath fidelity is validated implicitly: if interference effects were not captured, synthesized RF data would deviate significantly from real signals, leading to degraded quality metrics. The strong alignment of synthesized and measured signals thus confirms that multipath effects are effectively represented.

**Antenna Beam Patterns.** Directional antenna effects, including side lobes and attenuation, are incorporated in GSRF through data-driven learning rather than explicit physics-based parameterization. The Fourier–Legendre basis provides the representational capacity: Legendre polynomials capture polar variations such as main beam gain and off-axis attenuation, while Fourier components represent azimuthal phase shifts and side lobe structures. During rendering, these directional dependencies are aggregated via spherical ray tracing, and orthographic splatting ensures that beam-induced modulations are preserved in the synthesized RF field. If training data reflects beam-specific effects, the optimization naturally adapts Gaussian attributes to encode them.

In our datasets, the antennas used are omnidirectional, so side lobes and beam shaping are not observed. Nevertheless, ablation studies of the Fourier–Legendre basis confirm its benefit, showing improved performance even under isotropic conditions. We anticipate that with directional antenna data (*e.g.*, beamformed phased arrays), GSRF would capture and reproduce beam patterns faithfully, as the framework is agnostic to antenna type and adapts to observed propagation characteristics.

