# OpenReview forum: "GSRF: Complex-Valued 3D Gaussian Splatting for Efficient Radio-Frequency Data Synthesis"
_NeurIPS.cc/2025/Conference — NeurIPS 2025 spotlight_

### Official Review · Reviewer_w6X4 · 2025-06-25

**Clarity:** 4
**Significance:** 2
**Originality:** 2
**Rating:** 5
**Confidence:** 4

**Summary:**

This work presents a physically grounded extension of 3D Gaussian splatting (3DGS) for reconstructing RF signals, leveraging the Huygens-Fresnel principle to treat each Gaussian blob as a secondary RF source. The method introduces complex-valued 3D Gaussians with Fourier-Legendre radiance fields models to represent both amplitude and phase-dependent interactions. The approach achieves state-of-the-art accuracy across multiple RF synthesis tasks (e.g., spatial spectrum, CSI, RSSI) while offering significant gains in training and inference speed over NeRF-based baselines.

**Questions:**

- Why are there no comparisons to RF-3DGS?

- Are the comparisons to WRF-GS or WRF-GS+, how does the proposed deformability in WRF-GS+ affect the results?

- Given how "blobby" the reconstructed content is, I'm not sure that SSIM is a useful loss function here? Are there any ablations on its purpose / improvement?

- Eq 10-12, the 2D recieving plane is modeled with equirectangular projection, is there a strong motivation here for this (as it will inevitably stretch points close to the poles)? Would something like a cube-map (see: Greene, Ned. "Environment mapping and other applications of world projections.") be useful here?

**Ethical Concerns:**

["NO or VERY MINOR ethics concerns only"]

**Final Justification:**

In light of the additional comparisons and clarifying details provided in the rebuttal, I believe that this work should be accepted into the conference proceedings. It advanced the state-of-the-art in modeling radio-frequency signal distributions in a computationally tractable way, and provides clear avenues for follow-on domain-specific applications.

**Limitations:**

The limitations of static scene reconstruction are addressed in the work, I do not see any major potential negative societal impacts of the work.

**Quality:**

2

**Strengths And Weaknesses:**

**Strengths:**
Foremost I’d like to commend the writing and formatting of this work, as the sections are laid out in clear and logical order, with consistent mathematical notation following established conventions. Overall the work is easy to follow, with physically intuitive reasons given behind most model choices (e.g., Fourier-Legendre Expansion, orthographic splatting). It addresses a real and relevant signal modeling problem, as simulating dense, realistic RF signal distributions remains a highly non-trivial task. The qualitative and quantitative results are compelling, with improvements demonstrate over both NeRF2 and WRF-GS in reconstruction and training speed.

**Weaknesses:**
As this is not the first work using radiance fields for RF modeling, nor the first work to use 3DGS for RF, the niche for novelty is limited to specifically the importance of phase information for reconstruction (as expressed via the presented complex ray tracing algorithm). Given this I’m somewhat surprised the comparisons focus more on NeRF2 than the prior 3DGS works (with no results for RF-3DGS? and no WRF-GS results in Figure 2,6,9,10). This does not inspire confidence that the proposed approach is a strict improvement over other 3DGS approaches.

While the writing in the paper is clear, I can’t fully follow what is being illustrated in Figure 1. Expanding the figure description might help, as would incorporating mathematical notation from the work directly into the elements of the figure.

---

> ### Author Rebuttal · Authors · 2025-07-30
>
> ## **To Reviewer w6X4**
> Thank you for your thoughtful assessment and for recognizing the strengths of our paper. Below, we provide responses to your questions and concerns.
>
> ### **Q1 – Contribution**
> GSRF advances 3DGS into the RF domain via a unified, complex-valued modeling pipeline:
>
> **Scene Representation.** GSRF introduces a compact hybrid Fourier–Legendre Expansion (FLE) basis to jointly encode angular dependence and frequency-sensitive phase shifts. This representation preserves phase coherence, which is essential for capturing phase-driven constructive and destructive interference patterns modulating signal amplitude.
>
> **Rendering.** GSRF projects Gaussians onto a spherical domain using orthographic splatting and applies ray-tracing-based blending to integrate complex-valued radiance and transmittance along propagation paths. This design aligns with spherical RF antenna reception and models RF propagation through path integrals, with fully differentiable gradients derived for all parameters.
>
> **Optimization.** GSRF optimizes Gaussian parameters using RF-customized CUDA kernels for both forward and backward passes. Explicit gradient derivations ensure efficient updates through phase-aware components. A Fourier-domain loss is applied to preserve frequency-domain fidelity.
>
> ### **Q2 – Comparison to RF-3DGS[27]**
> RF-3DGS adopts a two-stage pipeline: first, an optical 3DGS is trained on co-located visual data (e.g., RGB images or LiDAR) to estimate Gaussian means, covariances, and opacities. These parameters are then fixed, and RF attributes (e.g., path loss) are learned using RF measurements, and uses spherical harmonics (SH) to encode directional information. Rendering relies on perspective projection and rasterization inherited from camera-based NeRFs, assuming camera-like intrinsics and extrinsics.
>
> In contrast, in GSRF: (i) learns all Gaussian attributes are learned from RF measurements via optimization and adaptive density control; (ii) handles directional encoding is handled by the FLE, which preserves amplitude–phase coupling effectively; (iii) performs rendering is performed through orthographic splatting and ray-tracing-based blending aligned with spherical RF reception, enabling fully differentiable propagation paths.
>
> **Exclusion of Comparison.** RF-3DGS is not included in our evaluations due to its reliance on co-located visual inputs. This makes it incompatible with our RF-only benchmarks, which lack RGB or LiDAR data, and renders it infeasible for real-world scenarios such as indoor sensing, 5G CSI prediction, or IoT deployments. Its dependence on visual access limits its practicality as a general-purpose RF modeling solution.
>
> ### **Q3 – Comparison to WRF-GS[28]**
> WRF-GS employs two NeRF-like MLPs to regress Gaussian attributes—the radiated signal and attenuation—using the Gaussian centers and transmitter coordinates as inputs. This MLP-centric design inherits NeRF's computational bottlenecks:
>
> (i) Dense querying of the MLPs for attribute prediction introduces inefficiency in both training, due to high parameter count and redundancy, and inference, as each Gaussian requires MLP evaluations.
>
> (ii) WRF-GS handles radiated signals of each Gaussian by separating them into real and imaginary components. While this approach maintains complex operations, it may not capture angular dependencies and frequency-sensitive phase shifts as directly, potentially limiting modeling of phase-driven interference that modulates RF signal amplitude.
>
> In contrast, GSRF represents complex-valued Gaussian primitives using the FLE, which jointly encodes directional dependencies and frequency-sensitive phase shifts in a compact, orthogonal representation. This structure preserves amplitude–phase coupling without the need for decomposition. GSRF optimizes these primitives end-to-end using RF-customized forward and backward CUDA kernels, enabling efficient gradient flow through the entire pipeline and avoiding extensive per-Gaussian MLP querying for attribute learning.
>
> ### **Q4 – Contemporaneous Work: WRF-GS+**
> WRF-GS+ is an extension of WRF-GS, released on arXiv after the NeurIPS 2025 contemporaneous work definition date (March 1st, 2025). It improves upon WRF-GS by introducing deformable Gaussians that decouple static components (such as path loss) from dynamic components (such as multipath), using learned offsets in signal strength, rotation, and scaling. This design enhances synthesis quality by adaptively reshaping Gaussians to better capture high-frequency multipath effects. It also reduces parameter redundancy and mitigates the inefficiencies of WRF-GS’s MLP-based attribute learning.
>
> We acknowledge the strength of this design, which offers a principled mechanism for modeling environmental variation without changing the core architecture. GSRF could potentially incorporate these deformable mechanisms in future work.
>
> However, the two approaches remain methodologically distinct. GSRF emphasizes a unified, complex-valued RF modeling pipeline. This results in a fully differentiable, MLP-free framework tailored for RF propagation modeling.
>
> ### **Q5 – Figures 2, 6, 9, 10 for WRF-GS**
> **Figure 2 (Qualitative Spectra Synthesis).** We present only NeRF² as a visual baseline in the manuscript. However, we have uploaded extensive qualitative results for WRF-GS, NeRF², and GSRF in the supplementary materials. We will include WRF-GS in Figure 2.
>
> **Figure 6 (Measurement Density).** We extend the measurement density analysis to include WRF-GS. WRF-GS and GSRF, trained with 0.8 measurements/ft³, achieve MSEs of 0.002659 ± 0.003560 and 0.002147 ± 0.003343, respectively. Both are comparable to NeRF²’s 0.002405 ± 0.003623, despite being trained with a higher density of 7.8 measurements/ft³. This benefit stems from WRF-GS’s use of 3DGS’s explicit representation, where Gaussian primitives offer greater representational power and flexibility, improving efficiency over NeRF-based volumetric sampling.
>
> **Figures 9 and 10 (BLE RSSI Prediction and Localization).** We add WRF-GS results to both tasks using the full training set. GSRF reduces RSSI error by 3.92% compared to WRF-GS due to its unified RF modeling. Localization errors remain similar across models due to the resilience of KNN. KNN selects the *k* nearest neighbors in the RSSI fingerprints and averages their positions. This averaging acts as a low-pass filter, mitigating the effects of synthesis noise.
> ||NeRF²|GSRF|WRF-GS|
> |-|-|-|-|
> |RSSI error (dBm)|6.091±5.427|4.094±3.908|4.261±3.943|
> |Localization error (m)|0.699±0.804|0.479±0.692|0.481±0.685|
>
> ### **Q6 – SSIM Loss**
> We agree that the inherent blob-like primitives in 3DGS can lead to smoother, less pixel-sharp outputs. However, SSIM remains valuable as it emphasizes structural and perceptual similarity across spatial patterns, rather than focusing solely on the pixel-wise errors emphasized by L1.
>
> In RF synthesis, where outputs like spatial spectrum are treated as image-like data (e.g., directional signal power), SSIM helps preserve meaningful directional spatial spectrum structures. This aligns with standard practice in 3DGS work (e.g., 3DGS and WRF-GS), which incorporate SSIM in their losses to enhance perceptual quality.
>
> Our new ablation experiments show that removing SSIM (relying on L1 plus Fourier loss) degrades RFID spatial spectrum synthesis PSNR by 0.73 dB (from 22.64 to 21.91 dB), confirming that SSIM refines structural details.
>
> ### **Q7 – Equirectangular Projection and Cube-Map**
> We appreciate the suggestion of cube-maps and agree that equirectangular projection introduces stretching near the poles, a known artifact in spherical mappings. However, our choice is motivated by several key factors:
>
> **Simplicity and Compatibility.** RF spatial spectra from antenna arrays are naturally parameterized by azimuth and elevation, which aligns with the latitude–longitude grid of equirectangular projection. This enables uniform angular sampling without additional remapping.
>
> **Limited Impact of Polar Distortion.** In practical RF antenna systems, high elevations (greater than 60°–70°) correspond to paths directed toward ceilings, floors, or the sky, where signals typically experience the following:
>
> *(i) Higher attenuation.* Materials like concrete and soil absorb or reflect signals with significant loss. Skyward paths (e.g., satellite) are line-of-sight but rare in indoor or occluded outdoor environments.
>
> *(ii) Fewer multipath components (MPCs).* Most useful MPCs arise from reflections and scattering at mid-elevations (10°–50°), typically from walls and objects at human height.
>
> We did consider cube-map projections, but we ultimately opted against them for two main reasons:
>
> **Seam Artifacts and Gradient Discontinuities.** Cube-maps split the sphere into six faces, introducing seams that disrupt gradient continuity. In our differentiable rendering pipeline, these seams can cause instability in backpropagation.
>
> **Non-uniform Sampling Misalignment.** Cube-maps exhibit higher sampling density at face centers and lower at edges, which conflicts with the uniform angular resolution used in RF datasets (e.g., spatial spectrum).
> ### **Q8 – Figure 1**
> The confusion mainly arises because the figure lacks an explicit workflow and does not follow a typical left-to-right layout. We will improve this in the revision.
>
> The scene is represented as a collection of Gaussian primitives. Each primitive has a mean position μ, covariance Σ, and complex-valued radiance ψ and transmittance ρ.
>
> The set of Gaussians evolves during training. The model uses gradient-based optimization to update their attributes and adapt density based on the loss computed between predictions and ground truth.
>
> On the right, rays γ are emitted from the receiver and traced through the scene. Gaussians are splatted onto a 2D angular grid using orthographic projection, and the received complex-valued signal is obtained by aggregating contributions along each ray.

---

> > ### Comment · Reviewer_w6X4 · 2025-08-04
> >
> > This rebuttal resolves my questions on RF-3DGS (though may be worth highlighting again in the text this difference between the proposed single stage and prior two-stage pipeline), as well as comparisons to WRF-GS/WRF-GS+.
> >
> > Re: SSIM, would also potentially check the performance of MSE + Fourier loss (as per "A comprehensive assessment of the structural similarity index" Dosselmann, Richard., Yang, Xue Dong). This is not of great importance for the acceptance of the work, but SSIM just strikes me as a somewhat unmatched loss function to the problem/data space given it's lack of spatial edges.
> >
> > Overall I would update my score to accept the work given these clarifcations / changes.

---

> > > ### Author Response · Authors · 2025-08-05
> > >
> > > Dear Reviewer w6X4,
> > >
> > > We sincerely thank the reviewer for recognizing our earlier rebuttal responses and for your willingness to update the score in favor of acceptance. We truly appreciate your thoughtful engagement with our work.
> > >
> > > Regarding your insightful comment on the suitability of SSIM as a loss function for our RF data space, and the reference to Dosselmann and Yang’s analysis [1], we fully agree that the perceptual justification for SSIM has limitations. Their study evaluates SSIM’s behavior relative to MSE, demonstrating that SSIM often behaves as a scaled version of MSE, particularly in cases where images exhibit low structural contrast or minimal edge content. This observation is supported by equal-value hypersphere visualizations and consistently high Spearman rank correlations (often exceeding 0.95), indicating that SSIM’s structural component may contribute minimally in such regimes.
> > >
> > > In our setting, although the spatial RF spectrum lacks the sharp edges characteristic of natural images, it nonetheless exhibits meaningful global spatial structure arising from underlying physical propagation phenomena. As illustrated in Figure 2 of the submitted manuscript, for example at Location 5, the ground-truth spatial spectrum (first row, last column) displays a prominent central lobe (presumably corresponding to the line-of-sight path), lateral lobes (likely resulting from multipath reflections due to environmental scatterers), and additional weaker regions (possibly associated with diffraction or secondary scattering). While these patterns may appear locally smooth due to the continuous nature of wave propagation, they encode semantically critical spatial relationships such as the directional clustering of paths and relative amplitude distributions.
> > >
> > > Although SSIM operates on local patches (typically 11×11 windows with Gaussian weighting), it computes not only local luminance (mean intensity) and contrast (variance), but also structural similarity through normalized covariance. This is formally captured in the structure term:
> > >
> > > $$
> > > s(x, y) = \frac{\sigma_{xy} + C_3}{\sigma_x \sigma_y + C_3},
> > > $$
> > >
> > > which quantifies pixel covariation and the orientation or consistency of signal variation across neighborhoods. These localized measurements are then aggregated across the entire image via mean SSIM, forming a global assessment of preserved spatial relationships. In the case of spatial RF spectra, globally consistent patterns such as central symmetry, aligned multipath lobes, or angular spread manifest as repeated or correlated local structures, for example, covariant intensity gradients within lobes. SSIM is inherently sensitive to such relational alignments, whereas MSE treats all deviations independently as squared errors and lacks the capacity to differentiate between structured distortion, such as shifted lobes, and unstructured noise, such as random fluctuations.
> > >
> > > Therefore, even though SSIM is computed locally, its patchwise structural sensitivity, when averaged across the spatial field, provides a mechanism for evaluating whether the spatial coherence of RF patterns is preserved. This makes SSIM technically more aligned with the spatial nature of the prediction task compared to MSE, which solely penalizes magnitude errors without considering spatial arrangement or structural consistency.
> > >
> > > We also appreciate the reviewer’s suggestion to consider a composite loss function such as MSE combined with Fourier loss. However, in accordance with NeurIPS policy restricting the inclusion of new experimental results at this discussion stage, we are not certain whether additional experiments are allowed. Instead, we would like to provide insight based on existing results from our previous rebuttal response to "Q6 – SSIM Loss".
> > >
> > > In that analysis, removing SSIM and relying solely on L1 plus Fourier loss led to a degradation in RFID spatial spectrum synthesis performance. One might interpret this performance gap as arising from SSIM’s similarity to MSE. However, given that the Fourier loss already captures global spectral consistency, the additional improvement from including SSIM suggests a non-trivial contribution from its structural perception component beyond simple pixel-wise similarity.
> > >
> > > [1] Dosselmann, Richard, and Xue Dong Yang. "A comprehensive assessment of the structural similarity index." Signal, Image and Video Processing 5.1 (2011): 81-91.

---

> > > > ### Comment · Reviewer_w6X4 · 2025-08-05
> > > >
> > > > Thank you for the detailed response, I have no further clarifying questions.

---

> > > > > ### Author Response · Authors · 2025-08-05
> > > > >
> > > > > Dear Reviewer w6X4,
> > > > >
> > > > > Thank you for your follow-up and for taking the time to engage in the discussion. We appreciate your thoughtful comments and are glad that our responses were able to address your concerns. Your feedback has been valuable in helping us improve the clarity and presentation of our work.
> > > > >
> > > > > Thank you again for your careful review and support.

---

### Official Review · Reviewer_v8Xu · 2025-07-01

**Clarity:** 3
**Significance:** 4
**Originality:** 3
**Rating:** 5
**Confidence:** 4

**Summary:**

The paper proposes a novel framework that extends 3D Gaussian Splatting (3DGS) to the domain of radio-frequency (RF) signal synthesis. Incorporating complex-valued 3D Gaussians, orthographic splatting, and an RF-customized CUDA-based complex ray tracing algorithm, the proposed model seems successfully develop RF-specific 3DGS, leading to efficient RF data synthesis. The paper addresses the limitations of previous NeRF-based models, particularly in terms of inefficiency in training and phase modeling deficiencies. The model is validated across diverse RF technologies (RFID, BLE, 5G, LoRa), and demonstrates superior synthesis accuracy and performance efficiency.

**Questions:**

1. Why is a one-degree angular resolution assumed, despite it being predefined by the number and configuration of Tx/Rx antennas in typical RF systems?
2. How does the model capture and validate the influence of multipath effects? Is it dependent on the explicit ray tracing process utilized in the model?
3. What mechanisms are in place to simulate or incorporate antenna beam patterns, including side lobes and directional attenuation, which are significant in RF synthesis accuracy?
4. Given the comparable performance to NeRF on the 5G CSI dataset as discussed in weaknesses of the paper, results can suggest inefficiencies in phase estimation, or limitations of complex rendering in capturing RF-specific nuances during simulation. Needs more discussion on it.

**Ethical Concerns:**

["NO or VERY MINOR ethics concerns only"]

**Final Justification:**

Based on the novelty of the work in the context of integrating 3DGS into RF modality as well as the authors' detailed feedback to resolve my previous issues (regarding downstream tasks and multipath effects), I believe the paper can be accepted to the conference.

**Limitations:**

yes

**Paper Formatting Concerns:**

Not founded

**Quality:**

3

**Strengths And Weaknesses:**

**Strengths**
- The method introduces a domain-specific adaptation of 3DGS to RF environments using complex-valued Gaussian primitives, integrating amplitude and phase characteristics through a Fourier-Legendre Expansion.
- The orthographic splatting technique and wavefront-based CUDA kernel design enable high-resolution angular modeling and real-time synthesis capabilities.
- Extensive benchmarks show that GSRF outperforms RF NeRF and other 3DGS-based models (e.g., WRF-GS) in both quality and efficiency across multiple RF modalities.
- The model significantly reduces training data requirements and inference latency, making it well-suited for practical applications in real-time wireless sensing.

**Weaknesses**
- While the paper highlights GSRF’s high fidelity in train-test settings, it does not sufficiently discuss or empirically validate its potential for downstream semi-supervised learning tasks in RF tasks, which is a major motivation of RF data synthesis outlined in the introduction.
- In the 5G CSI synthesis experiment, GSRF achieves similar performance to NeRF-based methods despite the use of complex rendering. This raises concerns about the practical impact of complex-valued modeling, i.e., the efficacy of phase estimation.
- The assumption of one-degree angular resolution, which is not always feasible given predefined Tx/Rx configurations, is not clearly justified.
- Important RF characteristics such as the modeling of multipath propagation, side-lobe effects, and antenna beam pattern-induced attenuation are not thoroughly analyzed or empirically validated.

---

> ### Author Rebuttal · Authors · 2025-07-30
>
> ## **To Reviewer v8Xu**
> Thank you for your thoughtful assessment and for recognizing the strengths of our paper. Below, we provide responses to your questions and concerns.
>
>
> ### **Q1 – Downstream Tasks**
> Thank you for your comment on the need for further discussion and empirical validation of GSRF's potential in downstream RF tasks, as motivated in the introduction. While our primary focus is high-fidelity synthesis, we do evaluate GSRF’s practical impact through application-specific experiments that go beyond standard metrics. Below, we summarize key results from relevant sections to highlight these benefits.
>
> **Section 5.3: BLE-Based Localization.** We apply GSRF-synthesized BLE fingerprints (RSSI readings per location) to a downstream localization task using a k-NN regressor. The database is constructed entirely from GSRF-synthesized signals, instead of real fingerprints. During testing, each real-collected fingerprint is used to query the synthesized database, retrieving the *k*-nearest locations and averaging them to estimate the position. This demonstrates that GSRF-generated fingerprints can directly support indoor positioning tasks, especially when exhaustive real-world data collection is impractical. The low localization error confirms that GSRF outputs are usable and accurate, reducing reliance on dense site surveys.
>
> **Appendix F.3: LoRa Gateway Coverage Estimation.** We use GSRF to synthesize LoRa RSSI maps for gateway coverage estimation. The resulting predictions achieve near-zero bit error rate (BER) across tested areas. Visual comparisons (Figure 13) show strong alignment with ground truth, enabling informed decisions on gateway placement. This validates GSRF’s role in practical deployment planning. For example, in domains such as precision agriculture or smart cities, sparse measurements can be extended into complete, actionable coverage maps.
>
> **Appendix F.5: Practical Benefits.** We summarize broader utility such as fast retraining (~10 minutes) for adapting to environmental changes (e.g., furniture rearrangement), and effective data augmentation for downstream models. For instance, using GSRF-synthesized fingerprints reduces training time for localization models by up to 50% while maintaining comparable accuracy to those trained on physics-based simulators. This shows how GSRF supports dynamic RF environments, enabling efficient updates without full re-measurement.
>
>
> ### **Q2 – Similar Performance of GSRF to NeRF² for CSI Synthesis**
> We appreciate your observation regarding the practical impact of our complex-valued modeling. While GSRF achieves comparable SNR to NeRF² [11], this does not undermine the efficacy of phase estimation in GSRF; rather, it highlights how our approach enables phase-aware synthesis more efficiently.
>
> To clarify, NeRF² [11] also models phase explicitly. It uses an MLP network to output complex-valued signals, where the MLP takes voxel coordinates and transmitter coordinates as input and regresses both amplitude and phase components of the RF field. This allows NeRF² to capture interference effects, but its volumetric querying along rays incurs high computational costs, limiting speed and scalability. As a result, it inherits the computation overhead typical of NeRF-based methods.
>
> In contrast, GSRF's complex-valued rendering integrates phase directly into Gaussian primitives via a Fourier–Legendre basis expansion, enabling amplitude–phase interactions without MLP overhead. This results in effective phase estimation while achieving faster training and inference times. The efficiency gain stems from our direct optimization of Gaussian attributes and custom CUDA kernels, which avoid the redundant computations of NeRF²'s MLP approach.
>
>
> ### **Q3 – One Degree Angular Resolution**
> Thank you for your comment on the angular resolution assumption. We agree that clearer justification is needed, particularly regarding feasibility with predefined Tx/Rx configurations, where hardware limits (e.g., antenna array size) may dictate coarser sampling. The one-degree resolution is not a rigid assumption but is chosen to align with the collected data's inherent resolution and standard RF practices, balancing fidelity, coverage, and efficiency. Below, we elaborate based on antenna types, since this directly influences the effective resolution.
>
> **For Multi-Antenna Configurations.** Our method does not impose a fixed prediction resolution; instead, it adapts to the antenna array's capabilities and the training data's angular granularity. In multi-antenna setups, the resolution is determined by the array's spatial sampling theorem, where the angular resolution scales with the number of elements. For example, in a 4×4 uniform rectangular array used in our RFID dataset, the system supports approximately 1° resolution using algorithms like MUSIC (Multiple Signal Classification). We align our synthesis resolution with the dataset's acquisition: the RFID spatial spectrum is measured at 1° intervals over azimuth and elevation, so we adopt this setting to maintain consistency with training data and avoid interpolation artifacts.
>
> However, we acknowledge that predefined Tx/Rx configurations may have lower resolution due to fewer antenna elements. For instance, a 2×2 array might only achieve 2–3° resolution. GSRF can flexibly accommodate such settings by training and evaluating at the available resolution without modification, since the orthographic splatting and loss functions operate on arbitrary ray grids. This makes our method hardware-agnostic and adaptable.
>
> **For Single-Antenna Configurations.** In single-antenna cases, the received signal is a scalar (power) aggregated omnidirectionally, per antenna theory and has no inherent angular resolution. Here, we discretize the spherical rendering at 1° to ensure thorough coverage of propagation paths while maintaining computational efficiency. Finer bins (e.g., 0.5°) increase the ray count without significant fidelity gains, while coarser bins (e.g., 5°) may miss interference patterns. Our 1° choice aligns with conventions in RF Computer-Aided Design (CAD) scene model-based simulation tools such as Wireless InSite and the MATLAB Ray Tracing toolbox.
>
>
> ### **Q4 – Multipath Effects**
> Thank you for your comment on how GSRF captures and validates multipath effects, and its dependency on ray tracing. We'll address this step by step, grounding our explanation in the model's design and empirical approach.
>
> **Design.** GSRF is designed to capture multipath effects by representing the RF scene as a collection of complex-valued 3D Gaussians, where each Gaussian serves as a primitive that approximates a propagation path or interaction point. This design is grounded in RF physics: multipath requires modeling multiple signal paths, each introducing amplitude attenuation and phase shifts. GSRF handles this using complex-valued radiance and transmittance attributes, encoded via a Fourier–Legendre basis that models directional and frequency-sensitive variations.
>
> The capture of multipath is inherently tied to our ray tracing process, which emits rays from the receiver across a spherical surface and aggregates the contributions of intersecting Gaussians. Transmittance encodes phase shifts and attenuation due to path length. Summing complex contributions allows for phase-based interference—constructive when the phase difference is near zero and destructive when near π. This process effectively discretizes the continuous wave propagation integral. Without ray tracing, these path-specific interactions would be lost, and aggregation would reduce to a naive amplitude blending, which is insufficient for RF modeling at centimeter-scale wavelengths.
>
> **Empirical Validation.** For validation, we rely on real-collected datasets, where explicit ground-truth for multipath effects is unavailable due to the aggregated nature of measurements. Instead, we implicitly validate through the synthesized RF data's overall quality: if multipath were not captured, interference patterns would be inaccurate, leading to poor fidelity metrics.
>
>
> ### **Q5 – Antenna Beam Patterns**
> Thank you for your comment on the mechanisms for incorporating antenna beam patterns, including side lobes and directional attenuation, in GSRF. As a data-driven method, GSRF learns these characteristics implicitly from the training data rather than modeling them via explicit physics-based rules, allowing flexibility across antenna types.
>
> **Incorporating Beam Patterns.** GSRF models directional antenna effects through the Fourier–Legendre basis for complex-valued radiance and transmittance. This basis encodes angular dependencies: Legendre polynomials handle polar variations in amplitude (e.g., main beam gain and off-axis attenuation), while Fourier components capture azimuthal phase shifts and side lobe structures. During rendering, these directional effects are aggregated via ray tracing on the spherical surface, where rays simulate beam orientations, and the orthographic splatting projects Gaussians to preserve pattern-induced modulations. If the training data reflects beam-specific effects, such as side lobes from directional antennas, the model’s optimization adapts the Gaussian attributes to capture them naturally.
>
> **Datasets.** In our datasets, the antennas used are omnidirectional or isotropic, and thus side lobes and directional attenuation are not prominent. As a result, GSRF does not explicitly exhibit such effects in results. However, ablations of the Fourier–Legendre basis show that directional encoding improves performance, confirming its role in capturing any present beam patterns. We anticipate that if directional antenna data (e.g., beamformed phased arrays) were used, GSRF would learn and express those patterns, as the framework is agnostic to antenna type and adapts to observed propagation.

---

> > ### Comment · Reviewer_v8Xu · 2025-08-05
> >
> > Thanks for detailed response and clarification on my previous concerns. Overall, the rebuttal resolves my questions specifically regarding down-stream tasks and multipath effects. Therefore, I'll keep my rating as accept.

---

> > > ### Author Response · Authors · 2025-08-05
> > >
> > > Dear Reviewer v8Xu,
> > >
> > > Thank you very much for your thoughtful follow-up and for taking the time to carefully consider our responses. We’re glad to hear that the clarifications addressed your concerns regarding downstream tasks and multipath effects. We truly appreciate your support and constructive feedback.

---

### Official Review · Reviewer_PLyQ · 2025-07-02

**Clarity:** 3
**Significance:** 3
**Originality:** 4
**Rating:** 5
**Confidence:** 2

**Summary:**

The authors introduce GSRF which combines 3D Gaussian splitting for the RF domain to synthesize new signals. While NerF-based methods can achieve high performance, they are computationally expensive and have high inference latency. GSRF uses a Fourier-Legendre basis to model directional radiance and orthographic splitting for intersections with spherical surfaces. They also customize CUDa kernels for complex-valued ray tracing. The authors evaluate their method on a variety of domains and demonstrate significant improvements in training, inference times, and sample efficiency compared to NeRF methods.

**Questions:**

- Could the authors explain a bit more about why the Fourier-Legendre basis was chosen over standard spherical harmonics? What does it contribute?
- Why was cube-based initialization chosen and how well does it work for spherical surfaces like RF signals?
- I can understand how GSRF is much faster to train and inference than NeRF^2, but how is it so much faster than WRF-GS? Is this mostly due to the custom CUDA kernels?

**Ethical Concerns:**

["NO or VERY MINOR ethics concerns only"]

**Limitations:**

Yes

**Quality:**

4

**Strengths And Weaknesses:**

## Strengths
- Although I am only somewhat familiar with RF, Gaussian splats, and NeRF, the paper is easy enough to follow overall. Some of the experiments were difficult to understand.
- Novel contribution of adapting 3DGS to RF domain and proposes methods to overcome limitations of current methods (RF-3DGS and WRF-GS).
- Empirical results seem quite strong: GSRF not only achieves lower MSE and higher PSNR over baselines, it is also much faster to train and inference than both a NeRF-based method and 3DGS-based method.

## Weaknesses:
- The authors mention this in their limitations and I agree: there is almost no spatial generalization for novel scenes. This method seems to require retraining (although it is fast to train) for each scene separately (please correct me if I'm wrong).

---

> ### Author Rebuttal · Authors · 2025-07-30
>
> ## **To Reviewer PLyQ**
> Thank you for your thoughtful assessment and for recognizing the strengths of our paper. Below, we provide responses to your questions and concerns.
>
>
> ### **Q1 – Spatial Generalization**
> Thank you for your comment and for highlighting this aspect in our limitations discussion. You are correct that GSRF, like many radiance field-based methods (including 3DGS and NeRF variants), is designed as a scene-specific model and does not inherently generalize to novel scenes without retraining. This stems from its core architecture: the 3D Gaussians are optimized to overfit the specific propagation environment captured in the training data, modeling unique multipath effects, interference patterns, and scene geometry (e.g., reflections off walls or objects in a given room). As a result, applying a trained model to a new scene (e.g., a different room layout) would yield poor synthesis fidelity, as the Gaussians would not align with the unseen propagation dynamics.
>
> That said, the fast training time you noted mitigates this limitation in practice. Retraining on new data for each scene is feasible and aligns with RF applications, where environments often change incrementally (e.g., furniture rearrangements) rather than requiring broad generalization across vastly different scenes. For scenarios demanding stronger generalization, future extensions could incorporate hybrid approaches, such as integrating scene-agnostic priors (e.g., physics-based simulations) or meta-learning, but these are beyond our current scope focused on high-fidelity, efficient per-scene synthesis.
>
> To enable spatial generalization while building on GSRF's framework, we propose actionable solutions: pretraining GSRF on diverse RF datasets and fine-tuning on target scenes, initializing Gaussians to preserve general priors. Incorporating conditional inputs like scene geometry embeddings via a lightweight MLP modulating attributes. Hybridizing with physics simulators (e.g., Wireless InSite [19]) to seed Gaussians along simulated paths for zero-shot inference on similar layouts.
>
>
> ### **Q2 – Why Fourier–Legendre Basis Instead of Spherical Harmonics**
> Thank you for your question on the choice of Fourier–Legendre Expansion (FLE) over Spherical Harmonics (SH) for modeling directional radiance in GSRF. Below, we provide a theoretical explanation for this design decision, grounded in the fundamental differences between RF signal propagation and visible light rendering, as well as the mathematical properties of the bases. This choice stems from the need to handle the unique challenges of RF domains—such as centimeter-scale wavelengths that cause pronounced phase-dependent interference and diffraction—which SH is ill-suited to capture efficiently.
>
> **Limitations of SH in the RF Domain.** SH forms an orthogonal basis on the sphere, commonly used in 3DGS for visible light to represent smooth, low-frequency view-dependent effects like shading and reflections. While effective for nanoscale visible light wavelengths, SH struggles in RF due to:
>
> *(i) Low-Frequency Bias.* SH excels at band-limited, diffuse functions but converges slowly for high-frequency or oscillatory patterns. RF signals, with wavelengths around centimeters (e.g., 915 MHz in our RFID dataset), exhibit sharp phase-dependent interference (constructive or destructive) and diffraction over propagation paths. These create high-frequency, non-smooth variations in the radiance field.
>
> *(ii) Phase Insensitivity.* Standard SH uses real coefficients, poorly modeling complex-valued RF fields where phase drives interference (e.g., destructive cancellation). RF's longer wavelengths amplify these effects over distances, unlike light's smoother amplitude-based aggregation.
>
> **Why FLE.** FLE is a hybrid basis combining Fourier series (for azimuthal/phase periodicity) with Legendre polynomials (for elevation dependency), tailored for complex-valued, directional RF radiance. We chose FLE for its theoretical advantages in RF:
>
> *(i) Frequency-Domain Suitability.* The Fourier component naturally models periodic phase shifts and interference patterns inherent to wave propagation. This aligns with RF's wave-like nature (diffraction, scattering), allowing efficient capture of oscillatory behaviors without requiring high degrees. Legendre polynomials provide orthogonal support on the sphere, and the hybrid enables better representation of anisotropic, phase-sensitive fields.
>
> *(ii) Complex-Valued Support.* Unlike real SH, FLE's complex coefficients directly encode amplitude and phase, crucial for RF's interference modeling (e.g., constructive amplification). This integrates seamlessly with our complex-valued Gaussians and ray-tracing.
>
> *(iii) Physical Alignment.* RF antennas collect over spherical regions, and FLE's polar–azimuthal decomposition matches this geometry, better than SH's global harmonics for localized multipath effects.
>
>
> ### **Q3 – Cube-Based Initialization**
> Thank you for your question on the cube-based initialization strategy in GSRF. The term may be slightly misleading—this is essentially a uniform strategy for initializing Gaussian primitives. Below, we explain the rationale for this choice and its effectiveness.
>
> **Why Uniform Initial Distribution.** The uniform initial distribution is selected to ensure comprehensive coverage of the entire 3D scene from the outset, which accelerates model convergence during training. In GSRF, the scene is represented by 3D Gaussians that model RF propagation paths. Starting with Gaussians distributed uniformly across the bounding scene volume (encompassing the transmitter, receiver, and environment) provides a structured, balanced initial representation. This avoids the pitfalls of sparse or uneven starting points, allowing the densification and pruning process to refine the model more efficiently. Compared to alternatives like random initialization, this strategy reduces early imbalances and speeds up convergence toward stable, high-fidelity synthesis. While gradient-based updating allows random initialization to achieve comparable spectrum synthesis fidelity, it requires longer training time: 0.59 hours for random initialization compared to 0.27 hours for uniform initial distribution for RFID spatial spectrum synthesis.
>
> This approach is inspired by efficient scene representations in 3DGS but adapted for RF's volumetric, wave-like nature, where paths span the full space rather than just surfaces. It prioritizes coverage over randomness to handle RF's complex, multipath environments without relying on prior geometric data, e.g., no Structure-from-Motion (SFM) points as in visual 3DGS.
>
> **Effectiveness for Spherical Surfaces Like RF Signals.** RF signals are inherently spherical, as they are collected over a spherical region centered at the antenna, with propagation effects radiating omnidirectionally. Despite the initial distribution's volumetric structure, it performs well by fully enclosing the spherical domain, ensuring Gaussians are initialized to capture radial paths without gaps. This initialization is only for the scene representation process, not related to the rendering process. The orthographic splatting and spherical rendering handle projection onto spherical coordinates during training and inference. The Gaussian shapes are then adapted by the adaptive density control (densification and pruning), which refines ellipsoid orientations and positions to fit the spherical, multipath nature of RF data effectively. This ensures robust performance without geometry-specific biases.
>
>
> ### **Q4 – Why GSRF (Ours) Is Faster Than WRF-GS [28]**
> Thank you for your question on the training and inference time comparison between GSRF and WRF-GS. While GSRF indeed benefits from custom CUDA kernels for acceleration, the speed gains over WRF-GS stem from a combination of architectural differences and optimizations tailored to RF synthesis. Below, we elaborate on the key factors.
>
> WRF-GS employs two NeRF-like MLPs to regress Gaussian attributes, using the Gaussian centers and transmitter coordinates as inputs. This MLP-centric design inherits NeRF's computational bottlenecks:
>
> (i) Dense querying of the MLPs for attribute prediction introduces inefficiency in both training, due to high parameter count and redundancy, and inference, as each Gaussian requires MLP evaluations.
>
> (ii) During optimization, backpropagation through the MLPs scales poorly with scene complexity, as gradients must flow through deep networks for every Gaussian update, leading to longer convergence times.
>
> (iii) Inference in WRF-GS involves re-evaluating MLPs per query (e.g., for novel transmitter positions), adding latency that increases with the number of Gaussians or rays.
>
> In contrast, GSRF eliminates MLPs entirely by directly optimizing per-Gaussian attributes as learnable parameters. This explicit representation enables direct gradient updates via simple operations like complex-valued blending in the rendering equation, reducing both parameter overhead and computation per iteration.

---

### Official Review · Reviewer_c9tJ · 2025-07-02

**Clarity:** 4
**Significance:** 3
**Originality:** 3
**Rating:** 5
**Confidence:** 4

**Summary:**

GSRF extends 3D Gaussian Splatting to the RF domain by modeling each Gaussian with complex radiance and transmittance parameters to capture signal amplitude and phase. It replaces spherical harmonics with a Fourier-Legendre expansion to represent directional, phase-dependent radiance. Ray-surface intersections are identified by orthographically splatting Gaussians onto a Ray Emitting Spherical Surface, reducing per-ray computation. A CUDA-based ray tracing algorithm aggregates complex-valued contributions along each ray using the Huygens–Fresnel principle in a differentiable framework. Evaluated on RFID, BLE, LoRa, and 5G datasets, GSRF speeds up training and inference while matching or outperforming NeRF-based and 3DGS-based RF synthesis methods.

**Questions:**

How sensitive are results to the Fourier-Legendre degree L or the angular resolution?

**Ethical Concerns:**

["NO or VERY MINOR ethics concerns only"]

**Final Justification:**

I appreciate the authors for the detailed response. I will maintain my positive rating.

**Limitations:**

The authors discussed the limitations of the paper at a high-level but didn't show more detailed analysis such as presenting failure cases.

**Quality:**

3

**Strengths And Weaknesses:**

+++ The paper proposes a solid method to model a novel application of neural rendering. Specifically on using 3D gaussian splitting for radio frequency data. The paper addresses relevant challenges like complex-value ray tracing, orthographic splitting and directional and phase modeling. It is a novel framework for a significant problem.

+++ The method is very well-written and clear to readers. The high-level logic is very consistent and the paragraphs are well-connected, following a clear logical order.

+++ The experiments are pretty solid, as the authors use run extensive experiments on several datasets and presented both qualitative and quantitative results. The authors included sufficient details but also keep the writing relatively concise and presented the experimental results very clearly.

+++ The experiments contain detailed analysis and ablation studies to show statistical significance and the efficacy of individual technical choices.

+++ The paper already included a well-documented github repo to reproduce the experiments in the paper.

--- All experiments assume static geometry; moving obstacles or dynamic environments require additional modeling or retraining and I imagine extending dynamic environments are pressing for real-world applications.

--- Performance depends on choices like number of Gaussians, basis degree and grid resolution, but the paper provides limited guidance on tuning these. It'll be great if the authors could provide some insights into whether the performance is sensitive to these values and if there are failure cases where the training is not stable.

---

> ### Author Rebuttal · Authors · 2025-07-30
>
> ## **To Reviewer c9tJ**
> Thank you for your thoughtful assessment and for recognizing the strengths of our paper. Below, we provide responses to your questions and concerns.
>
>
> ### **Q1 – Assumption of Static Scenarios**
> Thank you for your comment and for highlighting this limitation. We fully agree that extending the method to support moving obstacles or time-varying scenes is critical for enhancing real-world applicability. As noted in Section 6 (Discussion), the current approach is optimized for static environments. When the scene changes, such as due to moving obstacles, the model would require retraining. This limits its temporal adaptation.
>
> To address this in future work, we propose extending GSRF with a deformable 3DGS mechanism that supports dynamic RF scene rendering. The preliminary idea is to maintain a shared set of complex-valued 3D Gaussians that represent the baseline RF field and to introduce a lightweight deformation module that models time-varying changes efficiently. This would leverage GSRF's existing efficiency in rendering to enable real-time adaptation without full retraining.
>
> **Spatiotemporal Structure Encoder.** A spatiotemporal structure encoder captures both spatial RF interactions (such as reflection and diffraction) and temporal dynamics (such as obstacle motion) by decomposing the 4D space–time volume (x, y, z, t) into six compact 2D planes: (xy, yz, xz, xt, yt, zt). This decomposition is motivated by computational efficiency and locality preservation. Modeling the full 4D tensor directly would require R⁴ × C parameters, which is prohibitive for real-time applications. In contrast, the six-plane strategy reduces this to 6 × R² × C, significantly lowering memory and compute demands. It enables localized modeling of spatial features (e.g., reflections in xy, xz, yz) and temporal changes (e.g., motion in xt, yt, zt), allowing scalable and efficient querying on CUDA while preserving the multipath and interference effects crucial to RF modeling.
>
> **Multi-Head Decoder for Deformation.** A lightweight multi-head decoder, composed of small MLPs, would then predict per-Gaussian deformations in position, rotation, and scaling based on the features extracted by the encoder. The complex-valued attributes (e.g., radiance) remain unchanged to preserve phase-aware RF modeling. This setup enables each Gaussian to deform dynamically over time, allowing GSRF to simulate evolving RF paths and interference patterns without requiring a new set of Gaussians for each time step.
>
>
> ### **Q2 – Hyperparameter: Fourier–Legendre Expansion (FLE) Degree**
> Thank you for your comment on the FLE basis degree `L` in our GSRF framework. Below, we provide a brief analysis of its impact, supported by experimental results.
>
> **Analysis.** The FLE basis degree `L` controls the expressiveness of GSRF’s complex-valued 3D Gaussians in modeling phase-aware RF propagation effects such as interference and diffraction. Low degrees (`L = 1–2`) use fewer coefficients and capture only low-frequency angular components, which leads to underfitting of complex RF patterns. Moderate degrees (`L = 3–4`) better capture essential phase variations with a compact number of Fourier–Legendre coefficients. High degrees (`L ≥ 5`) risk overfitting to noise and introduce increased computational cost and numerical instability. This behavior is similar to how the degree of spherical harmonics (SH) controls fidelity in other 3DGS-based systems. We find that `L = 3` provides a good balance between modeling fidelity and efficiency.
>
> **Experimental Validation.** We use the sparse training data with 0.8 measurements/ft³ for spatial spectrum synthesis, as described in Section 5.1 (RFID Spatial Spectrum Synthesis). Our experiments (see table below) show that `L = 3` closely approaches the peak PSNR of `L = 4` while providing faster training and inference. Lower degrees (`L = 1` and `2`) underfit and yield lower PSNR, whereas a higher degree (`L = 5`) offers no clear accuracy gain and increases computational cost. We recommend `L = 3` as the default setting for its balanced trade-off between accuracy and efficiency.
> | Degree `L`              | 1     | 2     | 3     | 4     | 5     |
> |-------------------------|-------|-------|-------|-------|-------|
> | PSNR (dB)               | 16.49 | 17.73 | 18.67 | 18.78 | 18.21 |
> | Training time (minutes) | 13.84 | 15.09 | 16.21 | 19.26 | 24.72 |
> | Inference time (ms)     | 2.96  | 3.38  | 4.18  | 6.27  | 8.62  |
>
>
> ### **Q3 – Hyperparameter: Angular Resolution**
> Thank you for your comment on the angular resolution parameter. The one-degree resolution in our setting is not a rigid hyperparameter but is selected to match the inherent resolution of the collected data and to align with standard RF practices, balancing fidelity, coverage, and computational efficiency. Below, we elaborate based on antenna types, as these directly influence the effective angular resolution.
>
> **Multi-Antenna Arrays.** Our method does not impose a fixed angular resolution; instead, it adapts to the antenna array's capabilities and the training data's granularity. In multi-antenna setups, the resolution is governed by the array’s spatial sampling theorem, which approximately scales with the number of antenna elements. For instance, in our RFID dataset, a 4×4 uniform rectangular array supports around 1° precision using algorithms such as MUSIC (Multiple Signal Classification). We align the synthesis resolution with the measurement setup: since the RFID data was collected at 1° intervals over azimuth and elevation, we maintain this resolution for consistency and to avoid interpolation artifacts. If the measurement data had coarser resolution (e.g., 2° due to a smaller array), GSRF could be trained and evaluated at that resolution without modification, as the orthographic splatting and loss functions are resolution-agnostic and operate on arbitrary ray grids.
>
> **Single-Antenna Configurations.** In single-antenna cases (e.g., for RSSI synthesis), the received signal is a scalar power aggregated from all directions, per the antenna theory (no inherent angular resolution). Here, we discretize the spherical rendering at 1° to ensure coverage of propagation paths. Essentially, we could reduce the resolution to lower computation cost while preserving sufficient accuracy. Finer bins (e.g., 0.5°) increase ray count without proportional gains in fidelity for centimeter-wavelength RF. Coarser bins (e.g., 5°) risk missing details of propagation effects. Our 1° choice aligns with conventions in RF Computer-Aided Design (CAD) scene model-based simulation tools such as Wireless InSite and the MATLAB Ray Tracing toolbox.
>
> **Experimental Validation.** The table below shows the experiments for angular resolution in single-antenna RSSI synthesis, as described in Section 5.3 (BLE Real-Valued RSSI Synthesis). At 1°, with 360 × 90 = 32,400 rays, GSRF achieves the lowest RSSI error due to dense angular sampling. As the resolution coarsens, fidelity drops: 2° resolution yields slightly worse error with 180 × 45 = 8,100 rays, and 5° degrades further with 72 × 18 = 1,296 rays, due to the loss of directional detail. Training and inference time scale accordingly. Coarser resolutions reduce runtime but increase RSSI prediction errors. This tradeoff enables flexible tuning: the use of 1° resolution is suitable for precision-demanding tasks, while coarser settings are preferable in deployment scenarios where speed is critical.
> | Angular Resolution   | 1°     | 2°     | 5°     |
> |----------------------|--------|--------|--------|
> | RSSI error (dBm)     | 4.094  | 4.493  | 6.518  |
> | Training time (minutes) | 10.23  | 8.56   | 4.92   |
> | Inference time (ms)  | 1.76   | 0.94   | 0.17   |
>
>
> ### **Q4 – Hyperparameter: Number of Gaussians**
> Thank you for your comment on the number of Gaussians parameter. The number of Gaussians is not a fixed hyperparameter; as noted in Appendix B, which details the training procedure following the original 3DGS [7], it is dynamically optimized via densification and pruning during training. This adaptive process iteratively adds (densifies) new Gaussians in under-reconstructed regions while pruning redundant or low-contribution ones. This ensures the model efficiently captures the scene's RF propagation complexity without manual tuning of the number of Gaussians, balancing representation power and efficiency.

---

### Note · Authors · 2025-08-13

## **Dear Reviewers, ACs, and SACs**
We thank all reviewers for their constructive and insightful feedback.
### **1. Summary**
This paper introduces GSRF, a unified complex-valued modeling pipeline for efficient, high-fidelity Radio-Frequency (RF) signal synthesis, featuring (i) Scene Representation: Fourier–Legendre Expansion (FLE) to encode angular dependence and frequency-sensitive phase shifts, (ii) Rendering: orthographic splatting with complex-valued ray tracing to model RF propagation in spherical antenna reception setups, and (iii) Optimization: RF-customized CUDA kernels with explicit gradients and a Fourier-domain loss for fast, phase-aware training.
### **2. Strengths**
- **Strong motivation:** This work addresses key RF synthesis challenges, such as phase-dependent interactions and NeRF inefficiencies, targeting practical wireless sensing applications (c9tJ, PLyQ, v8Xu, w6X4).
- **Novel technical innovations:** The method extends 3DGS to RF with complex-valued Gaussians, a Fourier–Legendre basis for directional and phase modeling, orthographic splatting, and wavefront-based CUDA kernels for efficient ray tracing and real-time synthesis (c9tJ, PLyQ, v8Xu, w6X4).
- **Strong performance:** It achieves higher fidelity, greater efficiency, and improved sample efficiency compared to state-of-the-art baselines across RFID, BLE, LoRa, and 5G, supported by comprehensive ablation studies and both qualitative and quantitative evaluations (c9tJ, PLyQ, v8Xu, w6X4).

### **3. Concerns**
- **Temporal/Spatial generalization:** As acknowledged in our limitations section, we propose future work on deformable GSRF integrating a spatiotemporal encoder/decoder to enhance adaptability (c9tJ, PLyQ).
- **Downstream tasks:** Validation is demonstrated on BLE localization (low error via synthesized fingerprints) and LoRa gateway coverage estimation with a near-zero bit error rate (v8Xu).
- **Parameters:** The number of Gaussians is optimized dynamically without manual tuning. The impact of the FLE degree on performance is demonstrated in new experiments, and the 1° angular resolution assumption aligns with RF practice while adapting to hardware constraints (c9tJ, v8Xu).
- **Baselines:** RF-3DGS is excluded due to its reliance on visual data. WRF-GS’s MLP-based design yields lower fidelity and efficiency, whereas GSRF uses FLE with RF-customized CUDA kernels for end-to-end optimization, ensuring efficient gradient flow and avoiding costly per-Gaussian MLP queries (w6X4).

---

### Decision · Program_Chairs · 2025-09-17

**Decision:**

Accept (spotlight)

**Comment:**

The paper introduces GSRF, a complex-valued extension of 3D Gaussian Splatting for RF signal synthesis. Reviewers broadly praised the novelty of extending 3DGS to the RF domain, the contributions (Fourier-Legendre basis, orthographic splatting, CUDA kernels for complex ray tracing), and the strong empirical results across diverse RF tasks (RFID, BLE, LoRa, 5G). The clarity of presentation, reproducibility via a public repository, and detailed ablations were also noted as strong points.

Questions on Fourier–Legendre vs. spherical harmonics, cube-based initialization, angular resolution assumptions, multipath handling, and beam patterns were addressed thoroughly in rebuttal. Reviewers generally found these explanations satisfactory.

After rebuttal and discussion, all reviewers confirmed their positive recommendations, leading to consensus for acceptance.
Overall, reviewers agreed that the work is technically solid, addresses a significant problem, and advances the state of the art in RF signal modeling with a clear, efficient, and reproducible approach. The AC concurs with the consensus and recommends acceptance.